# Axioms for Learning from Pairwise Comparisons

**Ritesh Noothigattu**
Carnegie Mellon University
riteshn@cmu.edu

**Dominik Peters**
Harvard University
dpeters@seas.harvard.edu

**Ariel D. Procaccia**
Harvard University
arielpro@seas.harvard.edu

## Abstract

To be well-behaved, systems that process preference data must satisfy certain conditions identified by economic decision theory and by social choice theory. In ML, preferences and rankings are commonly learned by fitting a probabilistic model to noisy preference data. The behavior of this learning process from the view of economic theory has previously been studied for the case where the data consists of rankings. In practice, it is more common to have only pairwise comparison data, and the formal properties of the associated learning problem are more challenging to analyze. We show that a large class of random utility models (including the Thurstone–Mosteller Model), when estimated using the MLE, satisfy a Pareto efficiency condition. These models also satisfy a strong monotonicity property, which implies that the learning process is responsive to input data. On the other hand, we show that these models fail certain other consistency conditions from social choice theory, and in particular do not always follow the majority opinion. Our results inform existing and future applications of random utility models for societal decision making.

## 1 Introduction

More than two centuries ago, the marquis de Condorcet [1785] suggested a statistical interpretation of voting. Each vote, Condorcet argued, can be seen as a noisy estimate of a ground-truth ranking of the alternatives. A voting rule should aggregate votes into a ranking that is most likely to coincide with the ground truth. Although Condorcet put forward a specific noise model, his reasoning applies to any *random noise model*, which is a distribution over votes parameterized by a ground truth ranking [Conitzer and Sandholm, 2005, Caragiannis et al., 2016].

Until NeurIPS 2014, this statistical approach to voting was studied in parallel to the more common normative approach, which evaluates voting rules based on axiomatic properties. But the two approaches converged in a paper by Azari Soufiani et al. [2014], whose key idea was to determine whether maximum likelihood estimators (MLEs) for two noise models satisfy basic axiomatic properties. Their results were sharpened and extended by Xia [2016].

Our point of departure is that instead of random noise models we consider *random utility models*, where each alternative $x$ has a utility $\beta_x$, and the probability of drawing a pairwise comparison that puts $x$ above $y$ depends on $\beta_x$ and $\beta_y$. For example, under the well-known Thurstone–Mosteller Model [Thurstone, 1927, Mosteller, 1951], this pairwise comparison would be generated by sampling $u(x)$ and $u(y)$ from normal distributions with unit variance and means $\beta_x$ and $\beta_y$, respectively. Then $x$ is ranked above $y$ if and only if $u(x) > u(y)$.

Our research question, then, is this:

> *For a given random utility model, consider the aggregation rule that takes pairwise comparisons between alternatives as input and returns the ranking over alternatives defined by the MLE; which axioms does it satisfy?*

## 1.1 Why Is the Research Question Important?

The MLE, as an aggregation rule, is *statistically* well-motivated. From a voter's perspective, though, it is not immediately clear the the MLE is a good rule that adequately aggregates preferences. In particular, in case the statistical assumptions of a random utility fail to capture reality, the MLE may give a bad result. However, if we can show that the MLE satisfies standard axioms from voting theory, this implies a certain degree of robustness. It also provides reassurance that the statistical process cannot fall prey to pathological behavior in edge cases.

A string of recent papers [Freedman et al., 2020, Noothigattu et al., 2018, Kahng et al., 2019, Lee et al., 2019] proposes a sequence of systems for automated societal decision making through social choice and machine learning. They all aggregate pairwise comparisons, by fitting them to random utility models. As these systems are being deployed to support decisions in non-profits and government, it becomes crucial to understand normative properties of this framework.[1]

The work of Freedman et al. [2020] provides a concrete illustration. The paper deals with prioritization of patients in kidney exchange. They asked workers on Amazon Mechanical Turk to decide which of each given pair of patients with chronic kidney disease (defined by their medical profiles) should receive a kidney, and computed the MLE utilities assuming the pairwise comparisons were generated by a Bradley–Terry model [Bradley, 1984]. The resulting ranking over profiles was used to help a kidney exchange algorithm prioritize some patients over others. As is common, the authors pooled pairwise comparisons reported by many different voters, and fit a single random utility model to the pool, as opposed to fitting models to individual voters. This move is usually done to improve statistical accuracy, but, to an extent, it invalidates the underlying motivation of the noise model, which imagines a single decision maker with imprecise perception of utilities. Pooling assumes that a group of agents can be captured by the same model. Given this leap of faith, we believe that a normative analysis of the process becomes especially important.

Other papers apply an emerging approach called *virtual democracy* [Kahng et al., 2019] to automate decisions in two domains, autonomous vehicles [Noothigattu et al., 2018] and food allocation [Lee et al., 2019]. Lee et al. [2019] asked stakeholders in a nonprofit food rescue organization to report which of each given pair of recipient organizations should be allocated an incoming food donation. Unlike Freedman et al. [2020], they fit a random utility model (Thurstone–Mosteller) to the pairwise comparisons provided by each stakeholder *individually*, and used the Borda voting rule to aggregate the predictions given by each of the individual models. On the one hand, our axiomatic results may justify a move to the pooled approach of Freedman et al. [2020], which could improve accuracy. On the other hand, even when learning individual models, axiomatics can convince voters that their preferences are learned using a sensible method.

## 1.2 Our Results

We examine four axiomatic properties, suitably adapted to our setting. Informally, they are:

- *Pareto efficiency:* If $x$ dominates $y$ in the input dataset, $x$ should be above $y$ in the MLE ranking.

- *Monotonicity:* Adding $a \succ b$ comparisons to the input dataset can only help $a$ and harm $b$ in the MLE ranking.

- *Pairwise majority consistency:* If the input dataset is consistent with a ranking over the alternatives, that ranking must coincide with the MLE ranking.

- *Separability:* If $a$ is preferred to $b$ in the MLE rankings of two different datasets, $a$ must also be preferred to $b$ in the MLE ranking of the combined dataset.

The first two properties, Pareto efficiency and monotonicity, have immediate appeal and seem crucial: a system violating these is not faithful to input preferences. In Sections 3 and 4 we show that both properties are satisfied by a large class of random utility models when fitted using MLEs. For monotonicity, our main result, the proof is surprisingly involved, since we need to reason about the optimum utility values of all alternatives simultaneously. (In contrast, for random noise models, monotonicity is a simple consequence of the definition [Azari Soufiani et al., 2014, Xia, 2016].)

The latter two properties are *not* satisfied by MLEs, for all random utility models satisfying mild conditions. In a way, these negative results illuminate the behavior of random utility models: The case of pairwise majority consistency illustrates a trade-off, where random utility models ensure that a strong preference is respected, even if this leads them to override a majority preference elsewhere. While negative, we do not see the counterexamples in Sections 5 and 6 as pathological, though they may suggest contexts in which the use of random utility models is not appropriate.

## 2 Model

Let $\mathcal{X}$ be a finite set of alternatives. For notational simplicity, we let $\mathcal{X}^2 = \{(x, y) : x, y \in \mathcal{X}, x \neq y\}$ denote the set of *distinct* pairs of alternatives. Let $\# : \mathcal{X}^2 \to \mathbb{N}$ be a *dataset* of pairwise comparisons between alternatives: For $x, y \in \mathcal{X}$, $\#\{x \succ y\}$ is the number of times $x$ beat $y$ in the dataset.

The (pairwise) *comparison graph* $\mathcal{G}_\# = (\mathcal{X}, E)$ with respect to dataset $\#$ is the directed graph with the alternatives $\mathcal{X}$ as the vertices, and edges $E$ such that there exists a directed edge $(u, v) \in E$ iff $\#\{u \succ v\} > 0$. We say that $\mathcal{G}_\#$ is *connected* if its undirected form is connected, and we call it *strongly connected* if for all $(x, y) \in \mathcal{X}^2$, there is a directed path from $x$ to $y$ in $\mathcal{G}_\#$.

Given a dataset, our goal is to learn a random utility model (RUM). A random utility model specifies, for any two distinct alternatives $x, y \in \mathcal{X}$, the probability that when asking the decision maker to compare $x$ and $y$, the answer will be $x > y$. (Due to noise, when repeatedly querying the same pair, we may see different answers.) For us, a random utility model is parameterized by a vector $\beta \in \mathbb{R}^\mathcal{X}$, where $\beta_x$ is an unknown utility value for $x \in \mathcal{X}$. When we ask for a comparison between two alternatives $x, y \in \mathcal{X}$, we model the decision maker as sampling noisy utilities $u(x)$ and $u(y)$ from distributions parameterized by (and typically centered at) $\beta_x$ and $\beta_y$. Then, the decision maker reports the comparison $x > y$ iff $u(x) > u(y)$.

In this paper, we focus on random utility models with i.i.d. noise, so that $u(x) = \beta_x + \zeta(x)$, where $\zeta(x) \sim \mathcal{P}$ is i.i.d. across all alternatives. Let $F$ be the CDF of a random variable which is the difference between two independent random variables with distribution $\mathcal{P}$. Then the probability that alternative $x$ beats $y$ when they are compared is[2]

$$\Pr(x \succ y) = \Pr(u(x) > u(y)) = \Pr(\zeta(y) - \zeta(x) < \beta_x - \beta_y) = F(\beta_x - \beta_y). \quad (1)$$

We derived Equation (1) from a specific noise model, but it makes sense for any function $F : \mathbb{R} \to [0, 1]$ with CDF-like properties, even if it does not correspond to a noise distribution $\mathcal{P}$. Indeed, we can take any $F$ which is non-decreasing, satisfies $F(\Delta u) + F(-\Delta u) = 1$ for all $\Delta u \in \mathbb{R}$, and is such that $\lim_{\Delta u \to -\infty} F(\Delta u) = 0$ and $\lim_{\Delta u \to \infty} F(\Delta u) = 1$. We adopt Equation (1) as the general definition of a random utility model for our technical results.

Two of the most common random utility models are

- the *Thurstone–Mosteller (TM) model*: We sample utility as $u(x) = \beta_x + \zeta(x)$, with i.i.d. noise $\zeta(x) \sim \mathcal{N}(0, 1/2)$. This is equivalent to Equation (1) with $F$ as the Gaussian CDF $\Phi$.

- the *Bradley–Terry model* (equivalent to the *Plackett–Luce* model restricted to pairwise comparisons), where $\Pr(x \succ y) = \frac{e^{u(x)}}{e^{u(x)} + e^{u(y)}}$. This is Equation (1) with $F$ as the logistic function.

We usually assume that $F$ is strictly log-concave, and that it is strictly monotonic and continuous,[3] so that $F$ has an inverse on $(0, 1)$. These conditions hold for Thurstone–Mosteller and Bradley–Terry.

For a random utility model, given a dataset $\#$, our goal is to find parameters $(\beta_x)_{x \in \mathcal{X}}$ that best fit $\#$. We find these parameters by maximum likelihood estimation. The log-likelihood is given by

$$\mathcal{L}(\beta) = \sum_{(x,y) \in \mathcal{X}^2} \#\{x \succ y\} \log F(\beta_x - \beta_y).$$

When the dataset $\#$ is clear from the context, we write $\hat{\beta} \in \mathbb{R}^{\mathcal{X}}$ for a parameter vector that maximizes log-likelihood, and say that $\hat{\beta}$ is the *MLE*. Note that if $c \in \mathbb{R}$ is a scalar, then $\mathcal{L}(\beta) = \mathcal{L}(\beta + c)$ for all $\beta \in \mathbb{R}^{\mathcal{X}}$ (since $\Pr(x \succ y)$ depends only on the difference $\beta_x - \beta_y$), so the MLE is only defined up to an additive shift. For concreteness, we pick some $r \in \mathcal{X}$, call it the *reference alternative*, and fix $\beta_r = 0$; then, we maximize $\mathcal{L}$ over $\mathcal{D} = \{\beta \in \mathbb{R}^{\mathcal{X}} : \beta_r = 0\}$.

A random utility model is particularly appropriate when the dataset $\#$ consists of pairwise comparisons which are all reported by a single decision maker. However, in many cases the dataset is obtained by pooling reports from many agents, for instance to minimize the labeling effort of each individual agent, or if we have the explicit aim to aggregate preferences from different agents. Some of the axioms we study are explicitly motivated by cases where $\# = \sum_{i \in \mathcal{R}} \#_i$, i.e., the dataset is obtained by pooling individual datasets, where $\mathcal{R}$ is the set of agents. It then seems natural to assume that each agent behaves in accordance with some random utility model with unknown parameters $\beta^i$ and unknown CDF-like function $F_i$. Then the dataset $\#_i$ is generated by repeatedly querying the agent's random utility model for a comparison.

## 2.1 Existence and Boundedness of MLE

Before turning to our main results, we briefly state conditions that guarantee the existence of a finite MLE, and that guarantee uniqueness (up to a shift). Lemmas 2.1 and 2.4 have previously obtained in related contexts [Zermelo, 1929, Ford Jr, 1957, Zhao and Xia, 2018]; we include proofs to be self-contained.

In some scenarios, no finite $\beta$ maximizes likelihood, and thus the MLE may not exist. For instance, if some alternative $a$ beats other alternatives, but is not beaten even a single time in the dataset, the likelihood can always be strictly increased by increasing $\beta_a$ (when $F$ is strictly monotonic). Lemma 2.1 states a condition under which an MLE exists (i.e. $\mathcal{L}(\beta)$ has a maximizer). Its proof also provides a weak bound on one such maximizer. The proofs of the results in this section are in Appendix A.

**Lemma 2.1** (MLE exists). *Suppose $F$ is strictly monotonic and continuous. Then the MLE exists if and only if every connected component of the comparison graph $\mathcal{G}_{\#}$ is strongly connected.*

For alternative $x, y \in \mathcal{X}$, we define the *perfect-fit distance* between $x$ and $y$ as

$$\delta(x, y) := F^{-1}\left( \frac{\#\{x \succ y\}}{\#\{x \succ y\} + \#\{y \succ x\}} \right).$$

This is the difference in utilities of $x$ and $y$ required for the model to exactly match the observed frequencies of $\#\{x \succ y\}$ and $\#\{y \succ x\}$ in the data. We can check that the MLE will respect this perfect-fit distance when an alternative has only a single neighbor in the comparison graph.

**Lemma 2.2.** *Let $F$ be strictly monotonic and continuous. Suppose that for alternative $a$ there is exactly one alternative $b$ for which $\#\{a \succ b\} + \#\{b \succ a\} > 0$. If both $\#\{a \succ b\} > 0$ and $\#\{b \succ a\} > 0$, then any MLE $\hat{\beta}$ satisfies $\hat{\beta}_a - \hat{\beta}_b = \delta(a, b)$.*

We can use this result to provide a stronger bound on the MLE than the one from Lemma 2.1, which holds under slightly stronger conditions.

**Lemma 2.3.** *Suppose that $\#\{x \succ y\} > 0$ and $\#\{y \succ x\} > 0$ for all $x$ and $y$, and that $F$ is continuous and strictly monotonic. Then for every MLE $\hat{\beta}$ we have the bound*

$$\|\hat{\beta}\|_{\infty} \le |\mathcal{X}| \cdot \max_{(x,y) \in \mathcal{X}^2} \delta(x, y).$$

## 2.2 Uniqueness of MLE

Under mild conditions on the function $F$ and the comparison graph $\mathcal{G}_{\#}$, we have seen that a bounded MLE exists. When is the MLE unique? Note that if $F$ is a strictly log-concave, this implies that the

log-likelihood $\mathcal{L}(\beta)$ is concave. If we additionally require that the comparison graph $\mathcal{G}_{\#}$ is connected, then $\mathcal{L}(\beta)$ is in fact strictly concave, and thus the MLE is unique, as we prove in Appendix B.

**Lemma 2.4.** *Suppose that $F$ is strictly log-concave. Then $\mathcal{L}(\beta)$ is strictly concave and the MLE is unique (assuming it exists), if and only if the comparison graph $\mathcal{G}_{\#}$ is connected.*

# 3   Pareto Efficiency

A minimal requirement in economic theory is *Pareto efficiency*: if all agents prefer $a$ to $b$, then in aggregate, $a$ should be preferred to $b$. A first attempt at defining this notion for the environment of pairwise comparisons would be to say that if $\#\{a \succ b\} > 0$ but $\#\{b \succ a\} = 0$, then the MLE should satisfy $\hat{\beta}_a \geq \hat{\beta}_b$. However, this property is too restrictive. Consider a dataset with

$$\#\{a \succ b\} = 100, \#\{b \succ c\} = 1, \#\{c \succ a\} = 1,$$

and all other comparisons 0. To satisfy the mentioned property, the MLE would need to satisfy $\hat{\beta}_a \geq \hat{\beta}_b \geq \hat{\beta}_c \geq \hat{\beta}_a$, so they are all equal; however it seems better to have $\hat{\beta}_a > \hat{\beta}_b$.

A more sensible version of Pareto efficiency is motivated by the multi-agent setting described in Section 2, where $\# = \sum_{i \in \mathcal{R}} \#_i$, and each individual dataset $\#_i$ is generated by a random utility model with unknown parameters $\beta^i$. In this case, Pareto efficiency should say that if $\beta_a^i > \beta_b^i$ for all $i \in \mathcal{R}$, then the MLE $\hat{\beta}$ applied to dataset $\#$ should satisfy $\hat{\beta}_a > \hat{\beta}_b$ as well. Our official definition of Pareto efficiency implies this, but is phrased more generally.

**Definition 3.1** (Pareto efficiency). *Suppose $a, b \in \mathcal{X}$ satisfy $\#\{a \succ b\} > \#\{b \succ a\}$, and are such that for every other alternative $x \in \mathcal{X} \setminus \{a, b\}$, we have*

$$\#\{a \succ x\} > \#\{b \succ x\} \quad and \quad \#\{x \succ a\} < \#\{x \succ b\}.$$

*Then, for every MLE $\hat{\beta}$, Pareto efficiency requires that $\hat{\beta}_a \geq \hat{\beta}_b$.*

To see that this definition captures the desired behavior in the multi-agent case, note that if $\beta_a^i > \beta_b^i$, then the dataset $\#_i$ satisfies the condition of Definition 3.1 with high probability as we grow the number of comparisons in $\#_i$, and similarly the condition holds for the pooled dataset $\# = \sum_{i \in \mathcal{R}} \#_i$.

This version of Pareto efficiency is feasible; in fact, it is satisfied by most random utility models.

**Theorem 3.2.** *Maximum likelihood estimation satisfies Pareto efficiency if $F$ is strictly monotonic.*

The key idea behind the proof (given in Appendix C) is that if $a, b \in \mathcal{X}$ satisfy the condition of Definition 3.1 but some MLE $\hat{\beta}$ puts $\hat{\beta}_a < \hat{\beta}_b$, then the parameter vector $\beta$ equal to $\hat{\beta}$ except that $\beta_a = \hat{\beta}_b$ and $\beta_b = \hat{\beta}_a$ has strictly higher log-likelihood.

# 4   Monotonicity

If we add a pairwise comparison $a \succ b$ to a dataset, we should deduce that $a$ is stronger and $b$ is weaker relative to our previous estimates. It would be paradoxical if, upon seeing evidence that $a$ is strong and $b$ is weak, we decided to lower $a$'s utility or increase $b$'s utility. Monotonicity requires that this can never happen. We consider a strong form of this axiom, which requires that $a$ is strengthened relative to *every* other alternative, and not just relative to $b$.

**Definition 4.1** (Monotonicity). *Suppose that $\#$ and $\tilde{\#}$ are two datasets with unique MLEs $\hat{\beta}$ and $\tilde{\beta}$. Suppose that $\tilde{\#}\{x \succ y\} = \#\{x \succ y\}$ for all $x, y \in \mathcal{X}$ except that $\tilde{\#}\{a \succ b\} > \#\{a \succ b\}$. Then, monotonicity requires that for all $x \in \mathcal{X}$,*

$$\tilde{\beta}_a - \tilde{\beta}_x \geq \hat{\beta}_a - \hat{\beta}_x \quad and \quad \tilde{\beta}_b - \tilde{\beta}_x \leq \hat{\beta}_b - \hat{\beta}_x.$$

Equivalently, monotonicity requires that if $\#\{a \succ b\}$ *decreases*, then $a$ becomes weaker relative to other alternatives, and $b$ becomes stronger. We can interpret monotonicity as guaranteeing a kind of *participation incentive*: If we ask an agent to compare $a$ to $b$, the agent is assured that the answer can only influence our inferred utilities in the desired direction.

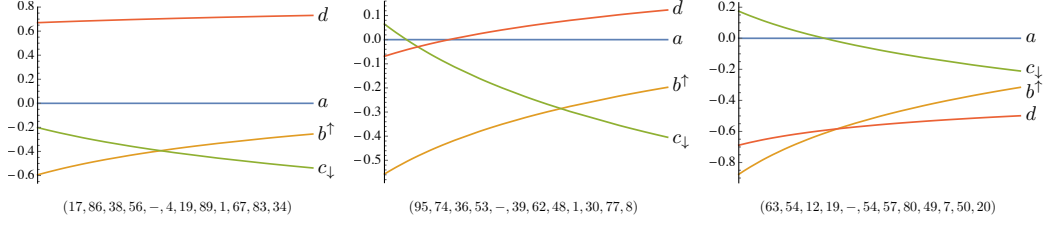

$(17, 86, 38, 56, -, 4, 19, 89, 1, 67, 83, 34)$      $(95, 74, 36, 53, -, 39, 62, 48, 1, 30, 77, 8)$      $(63, 54, 12, 19, -, 54, 57, 80, 49, 7, 50, 20)$

Figure 1: The MLE for Thurstone–Mosteller models is monotonic: with more $b \succ c$ comparisons, $b$'s utility increases, while $c$'s decreases. The vector shows the dataset $\#$ with $\mathcal{X}^2$ in lexic order, skipping comparisons of an alternative to itself: $(ab, ac, ad, ba, bc, bd, ca, cb, cd, da, db, dc)$.

Monotonicity is foundational to the idea of aggregating pairwise comparisons; in a sense, it encodes the proper meaning of a comparison "$a \succ b$". It may be surprising, then, that it is difficult to prove that MLEs of random utility models satisfy monotonicity.[4] While it is easy to check that the difference $\hat{\beta}_a - \hat{\beta}_b$ is increasing in $\#\{a \succ b\}$, it is much trickier to analyze the behavior of the log-likelihood for the positioning of alternatives other than $a$ and $b$. However, it turns out that random utility models do satisfy the strong monotonicity axiom. Our proof depends crucially on the assumption that $F$ is log-concave. Due to the conceptual importance of monotonicity, we consider this our main result.

**Theorem 4.2.** *Maximum likelihood estimation satisfies monotonicity if $F$ is strictly monotonic, log-concave, and differentiable.*

The proof, given in Appendix D, is relatively unwieldy. For intuition, let us provide an outline of a proof for the special case of three alternatives $a, b, c$. Let $\#$ and $\tilde{\#}$ be datasets that are identical except that $\#\{a \succ b\} < \tilde{\#}\{a \succ b\}$, and let $\hat{\beta}$ and $\tilde{\beta}$ be the respective MLEs, which are unique by Lemma 2.4. We take $a$ as reference, so $\hat{\beta}_a = \tilde{\beta}_a = 0$. It is easy to see that $\hat{\beta}_b \geq \tilde{\beta}_b$, since otherwise $\hat{\beta}$ would have greater log-likelihood than $\tilde{\beta}$ for the dataset $\tilde{\#}$, as $\hat{\beta}$ performs better on the $a$ vs $b$ comparisons, and performs no worse on other comparisons by optimality for $\#$. To see that also $\hat{\beta}_c \geq \tilde{\beta}_c$, consider first the dataset $\#$ and associated log-likelihood $\mathcal{L}(\beta_b, \beta_c)$ (with $\beta_a$ fixed to 0). Now, for $x \in \mathbb{R}$, let $\psi(x)$ denote the value of $\beta_c$ that maximizes $\mathcal{L}(x, \beta_c)$, i.e., maximizes likelihood among parameters $\beta$ with $\beta_a = 0$ and $\beta_b = x$. One can show that, since $F$ is strictly log-concave, $\psi(x)$ is increasing in $x$.[5] Notice that the number of comparisons between $a$ and $b$ in a dataset does not influence the optimum position of $\beta_c$, once $\beta_a$ and $\beta_b$ are fixed. Hence, the function $\psi$ is the same whether defined for $\#$ or for $\tilde{\#}$, since they only differ in $a$ vs $b$ comparisons. We have already seen that $\tilde{\beta}_b \leq \hat{\beta}_b$. Since $\psi$ is increasing, we have $\tilde{\beta}_c = \psi(\tilde{\beta}_b) \leq \psi(\hat{\beta}_b) = \hat{\beta}_c$, proving monotonicity.

To visualize monotonicity, consider the three examples in Figure 1. For four alternatives, we generated random datasets by choosing $\#\{x \succ y\}$ uniformly at random between 1 and 100, and picked three examples. In each case, we let $\#\{b \succ c\}$ vary from 0 to 100 (going horizontally from left to right), and show how the MLE of the Thurstone–Mosteller model changes as the number of $b \succ c$ comparisons grows; we fix $\hat{\beta}_a = 0$ as reference. As predicted by Theorem 4.2, the orange line of $\hat{\beta}_b$ is increasing, while the green line of $\hat{\beta}_c$ is decreasing. Note that the change in $\#\{b \succ c\}$ can affect other alternatives; in the middle figure, the relative positions of $a$ and $d$ swap.

In the pooled setting $\# = \sum_{i \in \mathcal{R}} \#_i$ of Section 2, where each agent $i \in \mathcal{R}$ is described by a random utility model with parameters $\beta^i$ that generates $\#_i$, a natural notion of monotonicity is this: Suppose we calculate the MLE $\hat{\beta}$ for $\#$ and suppose we increase the utility $\beta_a^i$ for some agent $i$ and some alternative $a$ while keeping all other parameters fixed. Then the updated MLE $\tilde{\beta}$ should satisfy $\tilde{\beta}_a - \tilde{\beta}_x \geq \hat{\beta}_a - \hat{\beta}_x$ for all $x \in \mathcal{X}$: the learned utility of $a$ increases relative to other alternatives.

Theorem 4.2 implies that random utility models (subject to the theorem's conditions) satisfy this pooled monotonicity notion with high probability, when $\#_i$ consists of many samples and when the number of comparisons is uniform across pairs. The reason is this: with high probability, the increase of $\beta_a^i$ increases the number of $a \succ x$ comparisons in $\#_i$ for all $x$. Assuming for now that no other dataset $\#_j$ and no other pairs in $\#_i$ are affected, then successively invoking Theorem 4.2 on $a \succ x$ pairs yields the result. Now, with some probability, other parts will be affected, but not too much. Since the MLE is continuous in $\#$ (see Appendix E), this noise will not invalidate monotonicity.

## 5 Pairwise Majority Consistency

Social choice theory has its root in the analysis of politics, where in many cases it is important to use aggregation rules that respect the wishes of a majority. A famous issue is that the "majority will" may not be coherent and in particular fail to be transitive. A minimal majoritarian requirement, thus, would be what we call *pairwise majority consistency (PMC)*: in cases where the majority produces a definite ranking, the aggregate should respect it.

**Definition 5.1.** *Suppose it is possible to label alternatives as $\mathcal{X} = \{x_1, \ldots, x_m\}$ such that whenever $i < j$, it holds that $\#\{x_i \succ x_j\} > \#\{x_j \succ x_i\}$. Then, pairwise majority consistency (PMC) requires that for every MLE $\hat{\beta}$, it holds that $\hat{\beta}_{x_i} \geq \hat{\beta}_{x_j}$ for all $i < j$.*

In contrast to our previous properties, PMC is violated by random utility models.

**Example 5.2.** Consider $\mathcal{X} = \{a, b, c\}$, and consider the dataset

$$\#\{a \succ b\} = 3, \#\{b \succ a\} = 2, \#\{a \succ c\} = 3, \#\{c \succ a\} = 2, \#\{b \succ c\} = 10, \#\{c \succ b\} = 1.$$

This dataset conforms to Definition 5.1 if we label $x_1, x_2, x_3 = a, b, c$. However, the unique MLE in the Thurstone–Mosteller model is $\hat{\beta}_a = 0$, $\hat{\beta}_b \approx 0.217$ and $\hat{\beta}_c \approx -0.751$, so that $\hat{\beta}_b > \hat{\beta}_a > \hat{\beta}_c$. The same example works for Bradley–Terry, which has MLE $\hat{\beta}_a = 0$, $\hat{\beta}_b \approx 0.316$ and $\hat{\beta}_c \approx -1.256$.

Why does the MLE not respect the majority ordering on this example? If the number $\#\{b \succ c\}$ was slightly above 1, we would obtain an MLE respecting the majority ordering, with $a \succ b \succ c$. However, as $\#\{b \succ c\}$ increases, due to the monotonicity of MLEs (Theorem 4.2), we find that $\hat{\beta}_b$ increases and $\hat{\beta}_c$ decreases. When $\#\{b \succ c\}$ becomes sufficiently large, $\hat{\beta}_b$ crosses $\hat{\beta}_a$. Thus, we find that the MLE has the 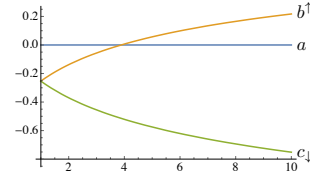 ordering $b \succ a \succ c$, which violates PMC. The figure on the right shows this behavior in the style of Figure 1, as $\#\{b \succ c\}$ increases from 1 to 10; we can see that PMC is violated from about 4.

This reasoning applies more generally to other random utility models beyond Thurstone–Mosteller, and we can construct similar counterexamples for a large class of such models; see Appendix E.

**Theorem 5.3.** *Maximum likelihood estimation violates pairwise majority consistency whenever $F$ is strictly monotonic, strictly log-concave, and differentiable.*

How frequent are PMC violations? Write $\%\{x \succ y\} = \#\{x \succ y\}/(\#\{x \succ y\} + \#\{y \succ x\})$ for the fraction of $x$ vs $y$ comparisons that $x$ wins. For $\mathcal{X} = \{a, b, c\}$, let $T$ be the space of datasets with

$$0.5 < \%\{a \succ b\}, \%\{a \succ c\}, \%\{b \succ c\} \leq 1.$$

For all datasets in $T$, PMC requires that $\hat{\beta}_a > \hat{\beta}_b > \hat{\beta}_c$. In Figure 2, we draw the cube $T$ and show the regions where the MLE for Thurstone–Mosteller fails PMC. Example 5.2, suitably normalized, falls in the upper orange region. Sampling uniformly over $T$, we find that Thurstone–Mosteller fails PMC in 17.8% of datasets, while Bradley–Terry fails in 16.6% of datasets.

## 6 Separability

We close by considering the *separability axiom* [Smith, 1973, Young, 1975]. It requires that when we merge two datasets, then wherever the MLE agreed on the datasets, this agreement is preserved in the combined dataset.

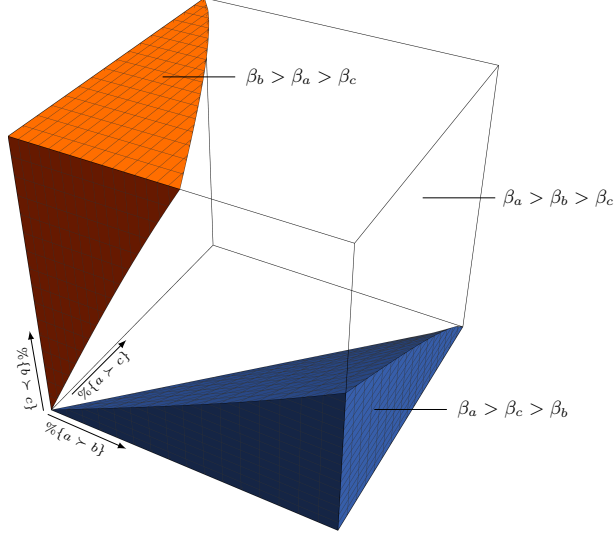

Figure 2: The cube shows all datasets in the space $T$, in which pairwise majority consistency requires that $\hat\beta_a > \hat\beta_b > \hat\beta_c$. The MLE for Thurstone-Mosteller models fails the condition in the shaded areas.

**Definition 6.1.** *Consider two datasets $\#^1$ and $\#^2$, and let $\hat\beta^1$ and $\hat\beta^2$ be MLEs. Suppose there exist two alternatives $a, b \in \mathcal{X}$ such that $\hat\beta_a^1 > \hat\beta_b^1$ and $\hat\beta_a^2 > \hat\beta_b^2$. Separability requires that for every MLE $\hat\beta$ for the pooled dataset $\# = \#^1 + \#^2$, it also holds that $\hat\beta_a > \hat\beta_b$.*

Separability is also called *consistency*, and seems particularly desirable in cases where we combine pairwise comparisons from different sources. While perhaps on first glance innocuous, separability is an extremely strong requirement, and few rules satisfy it; one can prove in general that separability constrains rules to be linear [Myerson, 1995]. Since likelihood maximization is not linear, it is no surprise that MLEs for random utility models fail separability.

**Example 6.2.** Let $\mathcal{X} = \{a, b, c\}$, and consider the two datasets

$$\#^1\{a \succ c\} = 6, \ \#^1\{c \succ a\} = 4, \ \#^1\{c \succ b\} = 100, \#^1\{b \succ c\} = 1, \text{ and}$$
$$\#^2\{a \succ c\} = 6, \ \#^2\{c \succ a\} = 4, \ \#^2\{b \succ a\} = 100, \#^2\{a \succ b\} = 1,$$

with 0 counts on all unspecified pairs. The unique MLEs for Thurstone–Mosteller on $\#^1$ and $\#^2$ are

$$\hat\beta_a^1 = 0, \hat\beta_b^1 \approx -2.58, \hat\beta_c^1 \approx -0.253; \quad \text{and} \quad \hat\beta_a^2 = 0, \hat\beta_b^2 \approx 2.330, \hat\beta_c^2 \approx -0.253.$$

We have both $\hat\beta_a^1 > \hat\beta_c^1$ and $\hat\beta_a^2 > \hat\beta_c^2$. However, the unique MLE on $\# = \#^1 + \#^2$ is $\hat\beta_a = 0$, $\hat\beta_b \approx 0.987$ and $\hat\beta_c \approx 1.973$. Thus. $\hat\beta_a < \hat\beta_c$, and so Thurstone–Mosteller violates separability. (The same example shows that Bradley–Terry violates separability.)

Intuitively, in both $\#_1$ and $\#_2$ there is a weak tendency to rank $a$ above $c$, and the MLE can implement this tendency without incurring any cost on other pairs (since Lemma 2.2 applies). However, once we combine the datasets, a strong consensus for $c \succ b \succ a$ emerges, and overriding this consensus to ensure $a \succ c$ is not worth it. While failing separability, the MLE's behavior seems perfectly sensible, and we prove in Appendix F that all random utility model do the same on this kind of example.

**Theorem 6.3.** *Maximum likelihood estimation violates separability whenever $F$ is strictly monotonic, strictly log-concave, and differentiable.*

Like for PMC, we can again ask on what percentage of (pairs of) datasets the MLE fails separately. Since we sample over pairs, we might guess the answer to be of lower order than in the case of PMC, and this is borne out by the data. For $m = 3$ alternatives, sampling uniformly over the space of datasets for which each pair of distinct alternatives is compared equally often, we find that on about 1.5% of dataset pairs, Thurstone–Mosteller fails separability. This fraction increases as $m$ increases, since there are more pairs of alternatives for which separability can be violated.

## 7  Discussion

To recap, we have established (under very mild assumptions) that the aggregation of pairwise comparisons via the MLE of a random utility model satisfies Pareto efficiency and monotonicity, and does not satisfy pairwise majority consistency and separability.

Our positive results deal with central properties that are required for an aggregation procedure: it does not override unanimous opinions (Pareto efficiency) and it incorporates new information (monotonicity). The latter property can be seen as a participation incentive, guaranteeing agents that each additional pairwise comparison will move the aggregate. Separability and pairwise majority consistency are not satisfied by random utility models, but arguably these properties are not as universally desirable. An analogy to the world of ranking-based voting rules is instructive, where separability characterizes a specific class of aggregators (positional scoring rules) [Young, 1975], but none of them satisfies pairwise majority consistency [Moulin, 1983].

Overall, we view our results as lending normative support to — and a more nuanced understanding of — existing and future applications of random utilities models for societal decision making.

## Broader Impact

Our work is motivated by the observation that the RUM-based approach for aggregating pairwise comparisons is already being considered for societal applications such as kidney exchange and food allocation. Our goal is to assess this choice from a normative viewpoint. Therefore, we view the potential implications of our work as being strictly positive. That said, one should keep in mind that learning-to-rank algorithms face a variety of ethical challenges, with fairness being one potential concern [Singh and Joachims, 2019, Beutel et al., 2019].

## Footnotes

[1]Previous work [Azari Soufiani et al., 2014, Xia, 2016] does not apply as it focuses on the aggregation of input *rankings* through random noise models.

[2] We assume $\mathcal{P}$ to be a continuous distribution, and so we do not have to worry about ties.

[3] Continuity of $F$ is guaranteed when the corresponding noise distribution $\mathcal{P}$ is continuous.

[4]For the Bradley–Terry model, monotonicity is easier to check, since the first-order conditions of likelihood maximization in that model are well-behaved [González-Díaz et al., 2014, Prop. 6.3]; that proof does not generalize to other models.

[5]Assume that $F$ is twice differentiable. Since $\log F$ is strictly concave, its second derivative is strictly negative. A straightforward calculation shows that then $\partial^2 \mathcal{L} / \partial \beta_c \partial \beta_c < 0$ and that $\partial^2 \mathcal{L} / \partial \beta_c \partial \beta_b > 0$. By definition of $\psi$, for each $x$, $(\partial \mathcal{L} / \partial \beta_c)(x, \psi(x)) = 0$. Since $\partial^2 \mathcal{L} / \partial \beta_c \partial \beta_b > 0$, the function $\partial \mathcal{L} / \partial \beta_c$ is increasing in its first argument, and so $(\partial \mathcal{L} / \partial \beta_c)(x + \Delta, \psi(x)) > 0$ for all $\Delta > 0$. Since $\partial^2 \mathcal{L} / \partial \beta_c \partial \beta_c < 0$, the function $\partial \mathcal{L} / \partial \beta_c$ is decreasing in its second argument, and hence $\psi(x + \Delta) > \psi(x)$, as desired.

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
