[Supplementary Material]

# A Appendix for Section 2.1

## A.1 Proof of Lemma 2.1

In this proof, we also show that under the given conditions, there exists an MLE $\hat{\beta}$ satisfying

$$\|\hat{\beta}\|_\infty \le -(|\mathcal{X}| - 1) \, F^{-1}\left(2^{-K/\eta}\right),$$

where $K = \sum_{(x,y)\in\mathcal{X}^2} \#\{x \succ y\}$ and $\eta = \min_{(x,y):\#\{x\succ y\}>0} \#\{x \succ y\}$.[6]

Suppose the comparison graph $\mathcal{G}_\#$ is such that each of its connected components is strongly connected. First, we show that moving any connected component (keeping all other distances fixed) does not change the likelihood. In particular, let $C$ be an arbitrary connected component that does not have the reference alternative $r$. The likelihood function can then be rewritten as

$$\mathcal{L}(\beta) = \sum_{x,y\in C} \#\{x \succ y\} \log F(\beta_x - \beta_y) + \sum_{x,y\notin C} \#\{x \succ y\} \log F(\beta_x - \beta_y),$$

as there are no edges between $C$ and its complement. For any vector $\beta \in \mathcal{D}$, define $\beta^\Delta \in \mathcal{D}$ for any $\Delta \in \mathbb{R}$ as follows

$$\beta_x^\Delta = \begin{cases} \beta_x + \Delta & \text{; if } x \in C \\ \beta_x & \text{; otherwise.} \end{cases}$$

That is, $\beta^\Delta$ is the same as $\beta$, except with utilities changed by the constant $\Delta$ for $C$. The likelihood at this point for any $\Delta$ is

$$\mathcal{L}(\beta^\Delta) = \sum_{x,y\in C} \#\{x \succ y\} \log F\left(\beta_x + \Delta - \beta_y - \Delta\right) + \sum_{x,y\notin C} \#\{x \succ y\} \log F(\beta_x - \beta_y) = \mathcal{L}(\beta).$$

Hence, adding any $\Delta$ to a connected component does not affect the likelihood. In particular, for any maximizer, we could set $\Delta$ such that an alternative (of choice) in the connected component $C$ has zero beta value, giving us a new maximizer. And this holds for every connected component. Hence, we just need to consider $\beta$ vectors which have a reference alternative in each of the connected components in order to find a maximizer. Let $r_1, r_2, \ldots, r_k$ denote the references we set in each of the connected components $C_1, C_2, \ldots, C_k$ respectively (where $k$ denotes the total number of connected components).

Define

$$B := -(|\mathcal{X}| - 1)F^{-1}\left(2^{-K/\eta}\right),$$

where $K$ and $\eta$ are as defined at the beginning of the proof. Consider an arbitrary beta vector (obeying the reference alternative constraints) with $\|\beta\|_\infty > B$. Then, there exists alternative $a \notin \{r_1, r_2, \ldots, r_k\}$ such that $|\beta_a| > B$. Without loss of generality, let $\beta_a > B$. Let $C_t$ be the connected component that $a$ lies in, with the reference alternative $r_t$. Consider all alternatives in $C_t$ whose $\beta$ value lies between that of $r_t$ and $a$. The total number of these alternatives (including the end points $r_t$ and $a$) is at most $|\mathcal{X}|$. Hence, the number of pairwise segments encountered starting from $r_t$ and ending at $a$ is at most $(|\mathcal{X}| - 1)$.[7] And since all these pairwise distances make up the total distance $\beta_a - \beta_{r_t} > B$, it implies that there exists at least one pairwise distance that is strictly larger than $B/(|\mathcal{X}| - 1)$. Let $(b, c)$ denote the ends of this pairwise segment. That is, $b, c \in C_t$ such that $\beta_c - \beta_b > \frac{B}{|\mathcal{X}|-1}$ and there is no alternative in $C_t$ with a $\beta$ value lying in the segment $(\beta_b, \beta_c)$. Let $\mathcal{U}$ denote the set of alternatives of $C_t$ that lie to the left of $b$, i.e., $\mathcal{U} = \{x \in C_t | \beta_x \le \beta_b\}$, and $\mathcal{V}$ be the set of alternatives of $C_t$ that lie to the right of $c$, i.e., $\mathcal{V} = \{x \in C_t | \beta_x \ge \beta_c\}$. Since no alternative in $C_t$ lies in between $b$ and $c$, $(\mathcal{U}, \mathcal{V})$ is a partition of $C_t$. Next, as every connected component is strongly connected, $C_t$ is also strongly connected. Hence, there has to be at least one edge going from $\mathcal{U}$ to $\mathcal{V}$ (otherwise, there would be no paths from alternatives in $\mathcal{U}$ to alternatives in $\mathcal{V}$ breaking strongly

connectedness). Let this edge be given by $(u, v) \in \mathcal{U} \times \mathcal{V}$. This implies that $\beta_u \leq \beta_b$, $\beta_v \geq \beta_c$ and $\#\{u \succ v\} > 0$. Hence, we have

$$\beta_v - \beta_u \geq \beta_c - \beta_b > \frac{B}{|\mathcal{X}| - 1}.$$

The log-likelihood can be rewritten as

$$
\begin{aligned}
\mathcal{L}(\beta) = \#\{u \succ v\} \log F(\beta_u - \beta_v) + \sum_{(x,y) \neq (u,v)} \#\{x \succ y\} \log F(\beta_x - \beta_y) \\
\leq \#\{u \succ v\} \log F(\beta_u - \beta_v) \\
< \#\{u \succ v\} \log F\left(-\frac{B}{|\mathcal{X}| - 1}\right),
\end{aligned}
\tag{2}
$$

where the first inequality holds because $\#\{x \succ y\} \geq 0$ and $\log F(\beta_x - \beta_y) \leq 0$ (as $F(\cdot) \leq 1$), and the second inequality holds because $\#\{u \succ v\} > 0$, $\beta_u - \beta_v < -\frac{B}{|\mathcal{X}|-1}$ and $\log F$ is strictly increasing. Next, consider the log-likelihood of the zero vector. We have,

$$\mathcal{L}(0) = \sum_{x \neq y} \#\{x \succ y\} \log F(0) = K \log F(0),$$

as $K$ is the total number of comparisons in the dataset. Recall the definition of $B$, we have,

$$B = -(|\mathcal{X}| - 1)F^{-1}\left(2^{-K/\eta}\right) \implies \log F\left(-\frac{B}{|\mathcal{X}| - 1}\right) = \frac{K}{\eta} \log\left(\frac{1}{2}\right).$$

Combining this with Equation (2), we have

$$
\begin{aligned}
\mathcal{L}(\beta) &< \#\{u \succ v\} \log F\left(-\frac{B}{|\mathcal{X}| - 1}\right) \\
&= \#\{u \succ v\} \frac{K}{\eta} \log\left(\frac{1}{2}\right) \\
&\leq K \log\left(\frac{1}{2}\right) \\
&= K \log F(0) = \mathcal{L}(0),
\end{aligned}
$$

where the inequality holds because $\eta = \min_{(x,y):\#\{x \succ y\}>0} \#\{x \succ y\} \leq \#\{u \succ v\}$ and $\log(1/2) < 0$, and the next equality holds as $F(0) = 1/2$. Hence, this shows that $\mathcal{L}(\beta) < \mathcal{L}(0)$ for any $\beta$ with $\|\beta\|_\infty > B$. In other words, such a $\beta$ vector cannot be a maximizer of $\mathcal{L}$. Therefore, to maximize $\mathcal{L}(\beta)$ we just need to consider $\beta$ vectors in $[-B, B]^{|\mathcal{X}|-k}$. And since this is a closed space, a maximizer always exists. Further, this maximizer satisfies $\|\hat{\beta}\|_\infty \leq B$.

Next, to prove the converse of the theorem statement, suppose there exists a connected component of $\mathcal{G}_\#$ that is not strongly connected. Denote this connected component by $C$. Consider all the strongly connected components of $C$; they form a DAG (as the condensation of a graph is always acyclic). Hence, there exists a strongly connected component in this DAG that has no incoming edge (from the rest of $C$). Let this strongly connected component be denoted by $S$. Further, as $C$ itself is a connected component, this implies that there exists at least one edge going from $S$ to $C \setminus S$. Putting all this together, we have strongly connected component $S$ such that there is no (incoming) edge from $\mathcal{X} \setminus S$ to $S$, and there is at least one (outgoing) edge from $S$ to $\mathcal{X} \setminus S$. Now, suppose for the sake of contradiction that $\mathcal{L}(\beta)$ has a maximizer. And, let $\hat{\beta}$ denote an MLE. The log-likelihood can be written as

$$
\begin{aligned}
\mathcal{L}(\beta) = \sum_{x,y \in S} \#\{x \succ y\} \log F(\beta_x - \beta_y) + \sum_{x \in S, y \notin S} \#\{x \succ y\} \log F(\beta_x - \beta_y) \\
+ \sum_{x,y \notin S} \#\{x \succ y\} \log F(\beta_x - \beta_y).
\end{aligned}
$$

Consider another beta vector $\tilde{\beta} \in \mathcal{D}$ that is the same as $\hat{\beta}$ except that it has beta values increased by a constant for alternatives in $S$.[8] For instance,

$$\tilde{\beta}_x = \begin{cases} \hat{\beta}_x + 1 & \text{; if } x \in S \\ \hat{\beta}_x & \text{; otherwise.} \end{cases}$$

The likelihood at this point is

$$\mathcal{L}(\tilde{\beta}) = \sum_{x,y \in S} \#\{x \succ y\} \log F(\hat{\beta}_x + 1 - \hat{\beta}_y - 1) + \sum_{x \in S, y \notin S} \#\{x \succ y\} \log F(\hat{\beta}_x - \hat{\beta}_y + 1)$$

$$+ \sum_{x,y \notin S} \#\{x \succ y\} \log F(\hat{\beta}_x - \hat{\beta}_y)$$

$$> \sum_{x,y \in S} \#\{x \succ y\} \log F(\hat{\beta}_x - \hat{\beta}_y) + \sum_{x \in S, y \notin S} \#\{x \succ y\} \log F(\hat{\beta}_x - \hat{\beta}_y)$$

$$+ \sum_{x,y \notin S} \#\{x \succ y\} \log F(\hat{\beta}_x - \hat{\beta}_y)$$

$$= \mathcal{L}(\hat{\beta}),$$

where the inequality holds because $\#\{x \succ y\} \geq 0$, $\log F(\hat{\beta}_x - \hat{\beta}_y + 1) > \log F(\hat{\beta}_x - \hat{\beta}_y)$ for all $(x, y) \in S \times S^C$ as $\log F$ is strictly increasing, and there exists at least one $(x, y) \in S \times S^C$ with $\#\{x \succ y\} > 0$ (because of the presence of the outgoing edge from $S$ to $S^C$). This leads to a contradiction as $\tilde{\beta}$ has strictly higher likelihood than the MLE $\hat{\beta}$. Therefore, an MLE does not exist. $\square$

### A.2 Proof of Lemma 2.2

Let $a$ be an alternative such that there is exactly one other alternative $b$ for which $\#\{a \succ b\} + \#\{b \succ a\} > 0$. The log-likelihood function is

$$\mathcal{L}(\beta) = \sum_{(x,y)} \#\{x \succ y\} \log F(\beta_x - \beta_y)$$

$$= \left[ \sum_{\substack{(x,y) \\ x \neq a, y \neq a}} \#\{x \succ y\} \log F(\beta_x - \beta_y) \right] + \#\{a \succ b\} \log F(\beta_a - \beta_b) + \#\{b \succ a\} \log F(\beta_y - \beta_x)$$

$$= \mathcal{G}(\beta_{-a}) + \#\{a \succ b\} \log F(\beta_a - \beta_b) + \#\{b \succ a\} \log F(\beta_b - \beta_a),$$

where $\mathcal{G}$ is the part of the likelihood function not containing $\beta_a$. Maximizing $\mathcal{L}(\beta)$ is equivalent to first maximizing with respect to $\beta_a$ and then with respect to the rest, $\beta_{-a}$.[9] Hence, we maximize

$$\#\{a \succ b\} \log F(\beta_a - \beta_b) + \#\{b \succ a\} \log F(\beta_b - \beta_a) \tag{3}$$

with respect to $\beta_a$.

**Claim A.1** (Coin flip likelihood). *For $h, t > 0$ and $p \in (0, 1)$, the function $f(p) = h \cdot log(p) + t \cdot log(1 - p)$ is strictly concave with the maximum uniquely attained at*

$$\hat{p} = \frac{h}{h + t}$$

*Proof.* $f'(p) = \frac{h}{p} - \frac{t}{1-p}$, and $f''(p) = -\frac{h}{p^2} - \frac{t}{(1-p)^2}$. Hence, $f''(p) < 0$ for all $p \in (0, 1)$ making $f$ a strictly concave function. Further, $f'(\hat{p}) = 0$. Hence, $\hat{p}$ as defined in the claim is the point where the maximum is attained. $\square$

Equation ($3$) can be rewritten as

$$\#\{a \succ b\} \log F(\beta_a - \beta_b) + \#\{b \succ a\} \log F(\beta_b - \beta_a) = f(F(\beta_a - \beta_b)),$$

where $f$ is the function from Claim A.1 with $h = \#\{a \succ b\} > 0$ and $t = \#\{b \succ a\} > 0$, as $F(\beta_b - \beta_a) = 1 - F(\beta_a - \beta_b)$. Applying Claim A.1, we have

$$f(F(\beta_a - \beta_b)) \leq f(\hat{p}),$$

for all $\beta_a, \beta_b$, where $\hat{p} = \frac{\#\{a \succ b\}}{\#\{a \succ b\} + \#\{b \succ a\}}$. Further, this upper bound can be achieved by setting $F(\beta_a - \beta_b) = \hat{p}$, which is possible as $F$ is invertible in $(0, 1)$ by strict monotonicty and continuity. Therefore, Equation ($3$) is uniquely maximized at $\beta_a = \beta_b + F^{-1}(\hat{p})$. And hence, every MLE satisfies

$$\hat{\beta}_a = \hat{\beta}_b + F^{-1}\left(\frac{\#\{a \succ b\}}{\#\{a \succ b\} + \#\{b \succ a\}}\right) = \hat{\beta}_b + \delta(a, b).$$

$\square$

## A.3  Proof of Lemma 2.3

The initial part of this proof is similar to the proof of Lemma 2.1. Let $B$ denote the bound $|\mathcal{X}| \cdot \max_{(x,y)} \delta(x, y)$. And, recall that $r$ denotes the alternative set as the reference, i.e. $\beta_r = 0$. Suppose for the sake of contradiction that there exists an MLE $\hat{\beta}$ with $\|\hat{\beta}\|_\infty > B$. This implies that there exists an alternative $a$ such that $|\hat{\beta}_a| > B$. WLOG, suppose $\hat{\beta}_a > B$. The number of alternatives whose $\beta$ value lies between that of $a$ and the reference $r$ (including both these points) is at most $|\mathcal{X}|$. Hence, the number of pairwise segments encountered starting from $r$ and ending at $a$ is at most $(|\mathcal{X}| - 1)$.[10] And since all these pairwise distances make up the total distance $\hat{\beta}_a - \hat{\beta}_r > B$, it implies that there exists at least one pairwise distance that is strictly larger than $B/(|\mathcal{X}| - 1)$. Let $(b, c)$ denote the ends of this pairwise segment. That is, $\hat{\beta}_c - \hat{\beta}_b > \frac{B}{|\mathcal{X}|-1}$, and there is no alternative with a $\beta$ value lying in the segment $(\hat{\beta}_b, \hat{\beta}_c)$. Construct a new beta vector $\tilde{\beta} \in \mathcal{D}$, such that $\tilde{\beta}$ is the same as $\hat{\beta}$ for alternatives to the left of alternative $b$, while is decreased by a small positive constant $\epsilon$ for all the other alternatives. That is,

$$\tilde{\beta}_x = \begin{cases} \hat{\beta}_x & ; \text{ if } \hat{\beta}_x \leq \hat{\beta}_b \\ \hat{\beta}_x - \epsilon & ; \text{ if } \hat{\beta}_x \geq \hat{\beta}_c. \end{cases}$$

In particular, choose $\epsilon$ such that the distance between $b$ and $c$ is still bigger than $\max_{(x,y)} \delta(x, y)$. This is possible because the original distance between $b$ and $c$ (i.e. $\hat{\beta}_c - \hat{\beta}_b$) is strictly larger than $\frac{B}{|\mathcal{X}|-1} = \frac{|\mathcal{X}|}{|\mathcal{X}|-1} \max_{(x,y)} \delta(x, y)$. Hence, one can choose $\epsilon > 0$ such that the new distance between $b$ and $c$ (i.e. $\tilde{\beta}_c - \tilde{\beta}_b$) is say the mid point of $\frac{|\mathcal{X}|}{|\mathcal{X}|-1} \max_{(x,y)} \delta(x, y)$ and $\max_{(x,y)} \delta(x, y)$. This would imply that we have

$$\tilde{\beta}_c - \tilde{\beta}_b > \max_{(x,y)} \delta(x, y). \tag{4}$$

Next, we show that in fact, $\mathcal{L}(\tilde{\beta}) > \mathcal{L}(\hat{\beta})$. The log-likelihood function is given as

$$\mathcal{L}(\beta) = \sum_{(x,y)\in\mathcal{X}^2} \#\{x \succ y\} \log F(\beta_x - \beta_y)$$

$$= \sum_{\{x,y\}\subseteq\mathcal{X}} \left[ \#\{x \succ y\} \log F(\beta_x - \beta_y) + \#\{y \succ x\} \log F(\beta_y - \beta_x) \right]$$

$$= \sum_{\{x,y\}\subseteq\mathcal{X}} f_{xy}(F(\beta_x - \beta_y)),$$

where $f_{xy}$ is the function from Claim A.1 with $h = \#\{x \succ y\} > 0$ and $t = \#\{y \succ x\} > 0$. Hence, from the claim, this function $f_{xy}$ is strictly concave with a maximum attained at $\hat{p}_{xy} =$

$\frac{\#\{x \succ y\}}{\#\{x \succ y\} + \#\{y \succ x\}}$. Let's call $\mathcal{U}$ as the set of alternatives $x$ with $\hat{\beta}_x \leq \hat{\beta}_b$ (i.e. the alternatives with $\beta$ value unchanged), and $\mathcal{V}$ as the set of alternatives $x$ with $\hat{\beta}_x \geq \hat{\beta}_c$ (i.e. the alternatives whose $\beta$ value is decreased by $\epsilon$). Observe that neither of these sets in empty, and they partition $\mathcal{X}$. Therefore, the log-likelihood at $\tilde{\beta}$ is

$$\mathcal{L}(\tilde{\beta}) = \sum_{\{x,y\} \subseteq \mathcal{X}} f_{xy}\left(F(\tilde{\beta}_x - \tilde{\beta}_y)\right)$$
$$= \sum_{\{x,y\} \subseteq \mathcal{U}} f_{xy}\left(F(\tilde{\beta}_x - \tilde{\beta}_y)\right) + \sum_{\{x,y\} \subseteq \mathcal{V}} f_{xy}\left(F(\tilde{\beta}_x - \tilde{\beta}_y)\right) + \sum_{(v,u) \in \mathcal{V} \times \mathcal{U}} f_{vu}\left(F(\tilde{\beta}_v - \tilde{\beta}_u)\right).$$

Note that, for $x, y \in \mathcal{U}$, the distance $(\tilde{\beta}_x - \tilde{\beta}_y)$ is the same as $(\hat{\beta}_x - \hat{\beta}_y)$ as the $\beta$ values are unchanged. In the case of $x, y \in \mathcal{V}$, again the distance $(\tilde{\beta}_x - \tilde{\beta}_y)$ is the same as $(\hat{\beta}_x - \hat{\beta}_y)$ as both $\beta$ values (of $x$ and $y$) are decreased by the same $\epsilon$. Finally, for any pair $(v, u) \in \mathcal{V} \times \mathcal{U}$, we have $\tilde{\beta}_v - \tilde{\beta}_u = \hat{\beta}_v - \hat{\beta}_u - \epsilon$, i.e. this pairwise distance decreases by $\epsilon$. Hence, the likelihood at $\tilde{\beta}$ becomes

$$\mathcal{L}(\tilde{\beta}) = \sum_{\{x,y\} \subseteq \mathcal{U}} f_{xy}\left(F(\hat{\beta}_x - \hat{\beta}_y)\right) + \sum_{\{x,y\} \subseteq \mathcal{V}} f_{xy}\left(F(\hat{\beta}_x - \hat{\beta}_y)\right) + \sum_{(v,u) \in \mathcal{V} \times \mathcal{U}} f_{vu}\left(F(\hat{\beta}_v - \hat{\beta}_u - \epsilon)\right).$$

Let's look at the terms $f_{vu}\left(F(\hat{\beta}_v - \hat{\beta}_u - \epsilon)\right)$ for $(v, u) \in \mathcal{V} \times \mathcal{U}$. We have

$$\hat{\beta}_v - \hat{\beta}_u > \hat{\beta}_v - \hat{\beta}_u - \epsilon = \tilde{\beta}_v - \tilde{\beta}_u \geq \tilde{\beta}_c - \tilde{\beta}_b > \max_{(x,y)} \delta(x, y) \geq \delta(v, u),$$

where the second inequality holds because $v \in \mathcal{V}$ is to the right of $c$ while $u \in \mathcal{U}$ is to the left of $b$, and the third inequality holds from Equation (4). Rewriting this equation keeping only the main components, we have

$$\hat{\beta}_v - \hat{\beta}_u > \tilde{\beta}_v - \tilde{\beta}_u > \delta(v, u).$$

As $F$ is a strictly increasing function, applying it to this equation gives us

$$F(\hat{\beta}_v - \hat{\beta}_u) > F(\tilde{\beta}_v - \tilde{\beta}_u) > F(\delta(v, u)) = \hat{p}_{vu},$$

where the equality holds by definition of the perfect-fit distance and $\hat{p}_{vu}$. Hence, by changing from $F(\hat{\beta}_v - \hat{\beta}_u)$ to $F(\tilde{\beta}_v - \tilde{\beta}_u)$, we move closer to the maxima of $f_{vu}$ (or alternatively, $F(\tilde{\beta}_v - \tilde{\beta}_u)$ is a convex combination of $F(\hat{\beta}_v - \hat{\beta}_u)$ and the maxima $\hat{p}_{vu}$). But, as $f_{vu}$ is strictly concave, it means that this change leads to an increase in its value. That is,

$$f_{vu}\left(F(\hat{\beta}_v - \hat{\beta}_u)\right) < f_{vu}\left(F(\tilde{\beta}_v - \tilde{\beta}_u)\right),$$

and this holds for every $(v, u) \in \mathcal{V} \times \mathcal{U}$. Hence, the log-likelihood at $\tilde{\beta}$ becomes

$$\mathcal{L}(\tilde{\beta}) = \sum_{\{x,y\} \subseteq \mathcal{U}} f_{xy}\left(F(\hat{\beta}_x - \hat{\beta}_y)\right) + \sum_{\{x,y\} \subseteq \mathcal{V}} f_{xy}\left(F(\hat{\beta}_x - \hat{\beta}_y)\right) + \sum_{(v,u) \in \mathcal{V} \times \mathcal{U}} f_{vu}\left(F(\tilde{\beta}_v - \tilde{\beta}_u)\right)$$
$$> \sum_{\{x,y\} \subseteq \mathcal{U}} f_{xy}\left(F(\hat{\beta}_x - \hat{\beta}_y)\right) + \sum_{\{x,y\} \subseteq \mathcal{V}} f_{xy}\left(F(\hat{\beta}_x - \hat{\beta}_y)\right) + \sum_{(v,u) \in \mathcal{V} \times \mathcal{U}} f_{vu}\left(F\left(\hat{\beta}_v - \hat{\beta}_u\right)\right)$$
$$= \mathcal{L}(\hat{\beta}).$$

That is, $\mathcal{L}(\tilde{\beta}) > \mathcal{L}(\hat{\beta})$, leading to a contradiction. Hence, for every MLE $\hat{\beta}$, we must have $\|\hat{\beta}\|_\infty \leq |\mathcal{X}| \cdot \max_{(x,y)} \delta(x, y)$. $\square$

# B  Proof of Lemma 2.4

The log-likelihood function is given as

$$\mathcal{L}(\beta) = \sum_{(x,y) \in \mathcal{X}^2} \#\{x \succ y\} \log F(\beta_x - \beta_y).$$

Consider $\beta \neq \gamma \in \mathcal{D}$ and $\theta \in (0, 1)$. Then,

$$\mathcal{L}(\theta\beta + (1-\theta)\gamma) = \sum_{(x,y)} \#\{x \succ y\} \log F(\theta\beta_x + (1-\theta)\gamma_x - \theta\beta_y - (1-\theta)\gamma_y)$$

$$= \sum_{(x,y)} \#\{x \succ y\} \log F(\theta(\beta_x - \beta_y) + (1-\theta)(\gamma_x - \gamma_y))$$

$$\geq \sum_{(x,y)} \#\{x \succ y\} \big[\theta \log F(\beta_x - \beta_y) + (1-\theta) \log F(\gamma_x - \gamma_y)\big]$$

$$= \theta \sum_{(x,y)} \#\{x \succ y\} \log F(\beta_x - \beta_y) + (1-\theta) \sum_{(x,y)} \#\{x \succ y\} \log F(\gamma_x - \gamma_y)$$

$$= \theta\mathcal{L}(\beta) + (1-\theta)\mathcal{L}(\gamma),$$

where the inequality holds because $\log F$ is concave, and $\#\{x \succ y\} \geq 0$ for every $(x, y) \in \mathcal{X}^2$. Hence, $\mathcal{L}$ is a concave function.

Next, suppose the comparison graph $\mathcal{G}_{\#}$ is connected. Recall, $r$ denotes the reference alternative set to zero. As $\beta \neq \gamma$, this implies that there exists an alternative $a \neq r$ such that $\beta_a \neq \gamma_a$. We know that the graph $\mathcal{G}_{\#}$ is connected, hence, there exists an undirected path from $a$ to $r$ in $\mathcal{G}_{\#}$. Let this (undirected) path be given as

$$a = v_0 \to v_1 \to v_2 \to \cdots \to v_t \to v_{t+1} = r.$$

As $\beta_a - \beta_r \neq \gamma_a - \gamma_r$, this implies that there exists $(l, l+1)$ such that $\beta_{v_l} - \beta_{v_{l+1}} \neq \gamma_{v_l} - \gamma_{v_{l+1}}$. Because if this difference was equal for all $l \in [0, t]$, it would imply that $\beta_a - \beta_r = \gamma_a - \gamma_r$. As there's an edge between $v_l$ and $v_{l+1}$, it implies that either $\#\{v_l \succ v_{l+1}\} > 0$ or $\#\{v_{l+1} \succ v_l\} > 0$. Without loss of generality, let $\#\{v_l \succ v_{l+1}\} > 0$. The log-likelihood is then

$$\mathcal{L}(\theta\beta + (1-\theta)\gamma) = \#\{v_l \succ v_{l+1}\} \log F(\theta(\beta_{v_l} - \beta_{v_{l+1}}) + (1-\theta)(\gamma_{v_l} - \gamma_{v_{l+1}}))$$

$$+ \sum_{(x,y) \neq (v_l, v_{l+1})} \#\{x \succ y\} \log F(\theta(\beta_x - \beta_y) + (1-\theta)(\gamma_x - \gamma_y))$$

$$> \#\{v_l \succ v_{l+1}\} \big[\theta \log F(\beta_{v_l} - \beta_{v_{l+1}}) + (1-\theta) \log F(\gamma_{v_l} - \gamma_{v_{l+1}})\big]$$

$$+ \sum_{(x,y) \neq (v_l, v_{l+1})} \#\{x \succ y\} \big[\theta \log F(\beta_x - \beta_y) + (1-\theta) \log F(\gamma_x - \gamma_y)\big]$$

$$= \theta\mathcal{L}(\beta) + (1-\theta)\mathcal{L}(\gamma)$$

where the strict inequality holds because $\#\{v_l \succ v_{l+1}\} > 0$, $\theta \in (0, 1)$, $\beta_{v_l} - \beta_{v_{l+1}} \neq \gamma_{v_l} - \gamma_{v_{l+1}}$ and $\log F$ is strictly concave. Therefore, $\mathcal{L}$ is strictly concave, and, it has unique maximizers.

For the converse, suppose the comparison graph $\mathcal{G}_{\#}$ is not connected (in the undirected form). As there is only one reference alternative $r$, let $C$ be a connected component that does not contain $r$. The log-likelihood can then be rewritten as

$$\mathcal{L}(\beta) = \sum_{x,y \in C} \#\{x \succ y\} \log F(\beta_x - \beta_y) + \sum_{x,y \notin C} \#\{x \succ y\} \log F(\beta_x - \beta_y),$$

as there are no edges between $C$ and its complement. Similar to proof of Lemma 2.1, for any vector $\beta \in \mathcal{D}$, define $\beta^{\Delta} \in \mathcal{D}$ for any $\Delta > 0$ as follows

$$\beta_z^{\Delta} = \begin{cases} \beta_z + \Delta & \text{; if } z \in C \\ \beta_z & \text{; if } z \notin C. \end{cases}$$

The likelihood at this point for any $\Delta$ is

$$\mathcal{L}(\beta^{\Delta}) = \sum_{x,y \in C} \#\{x \succ y\} \log F(\beta_x + \Delta - \beta_y - \Delta) + \sum_{x,y \notin C} \#\{x \succ y\} \log F(\beta_x - \beta_y) = \mathcal{L}(\beta).$$

$$(5)$$

Consider any $\theta \in (0, 1)$. Then,

$$(\theta\beta^{\Delta} + (1-\theta)\beta)_z = \begin{cases} \theta(\beta_z + \Delta) + (1-\theta)\beta_z = \beta_z + \theta\Delta & \text{; if } z \in C \\ \theta\beta_z + (1-\theta)\beta_z = \beta_z & \text{; if } z \notin C, \end{cases}$$

and hence implying that $\theta\beta^\Delta + (1-\theta)\beta = \beta^{\theta\Delta}$. In particular, this gives us

$$\mathcal{L}(\theta\beta^\Delta + (1-\theta)\beta) = \mathcal{L}(\beta^{\theta\Delta}) = \mathcal{L}(\beta) = \theta\mathcal{L}(\beta^\Delta) + (1-\theta)\mathcal{L}(\beta),$$

where the second equality holds because Equation (5) holds for any $\Delta > 0$ (including $\theta\Delta$). But, as $\beta^\Delta \neq \beta$ and $\theta \in (0,1)$, this implies that $\mathcal{L}$ is not strictly concave. Note that, this also shows that if an MLE $\hat\beta$ existed, it would not be unique. As, $\hat\beta^\Delta$, with say $\Delta = 1$, would have the same likelihood as $\hat\beta$ making it an MLE as well.

Hence, concluding the proof that $\mathcal{L}(\beta)$ is strictly concave and the MLE is unique, iff the comparison graph $\mathcal{G}_\#$ is connected. $\qquad\square$

## C  Proof of Theorem 3.2

Suppose the dataset is such that it satisfies the properties given in Definition 3.1, i.e., $\#\{a \succ b\} > \#\{b \succ a\}$, and for every other alternative $x \in \mathcal{X} \setminus \{a, b\}$, we have

$$\#\{a \succ x\} > \#\{b \succ x\} \quad \text{and} \quad \#\{x \succ a\} < \#\{x \succ b\}.$$

Suppose for the sake of contradiction that there exists an MLE $\hat\beta$ such that $\hat\beta_a < \hat\beta_b$. Construct $\tilde\beta$ such that it is the same as $\hat\beta$, except with $a$'s and $b$'s utilities swapped.[11] That is,

$$\tilde\beta_x = \begin{cases} \hat\beta_x; & \text{if } x \notin \{a, b\} \\ \hat\beta_b; & \text{if } x = a \\ \hat\beta_a; & \text{if } x = b. \end{cases}$$

The log-likelihood at the MLE $\hat\beta$ is given as

$$\begin{aligned}
\mathcal{L}(\beta) &= \sum_{(x,y)\in\mathcal{X}^2} \#\{x \succ y\} \log F(\beta_x - \beta_y) \\
&= \sum_{x,y \notin \{a,b\}} \#\{x \succ y\} \log F(\hat\beta_x - \hat\beta_y) \\
&\quad + \sum_{y \notin \{a,b\}} \#\{a \succ y\} \log F(\hat\beta_a - \hat\beta_y) + \sum_{y \notin \{a,b\}} \#\{b \succ y\} \log F(\hat\beta_b - \hat\beta_y) \\
&\quad + \sum_{x \notin \{a,b\}} \#\{x \succ a\} \log F(\hat\beta_x - \hat\beta_a) + \sum_{x \notin \{a,b\}} \#\{x \succ b\} \log F(\hat\beta_x - \hat\beta_b) \\
&\quad + \#\{a \succ b\} \log F(\hat\beta_a - \hat\beta_b) + \#\{b \succ a\} \log F(\hat\beta_b - \hat\beta_a).
\end{aligned} \tag{6}$$

Before proceeding with the proof, we prove a simple claim.

**Claim C.1.** *Let $c, d, e, f > 0$ such that $c > d$ and $e > f$. Then $ce + df > cf + de$.*

*Proof of Claim C.1.*

$$\begin{aligned}
ce + df &= c(f + (e - f)) + df \\
&= cf + c(e - f) + df \\
&> cf + d(e - f) + df \\
&= cf + de,
\end{aligned}$$

where the inequality holds because $c > d$ and $(e - f) > 0$. $\qquad\square$

By Claim C.1, for any $x, y \in \mathcal{X}$, we have,

$$\#\{a \succ y\} \log F(\hat{\beta}_a - \hat{\beta}_y) + \#\{b \succ y\} \log F(\hat{\beta}_b - \hat{\beta}_y)$$
$$< \#\{a \succ y\} \log F(\hat{\beta}_b - \hat{\beta}_y) + \#\{b \succ y\} \log F(\hat{\beta}_a - \hat{\beta}_y),$$

$$\#\{x \succ a\} \log F(\hat{\beta}_x - \hat{\beta}_a) + \#\{x \succ b\} \log F(\hat{\beta}_x - \hat{\beta}_b)$$
$$< \#\{x \succ b\} \log F(\hat{\beta}_x - \hat{\beta}_a) + \#\{x \succ a\} \log F(\hat{\beta}_x - \hat{\beta}_b),$$

$$\#\{a \succ b\} \log F(\hat{\beta}_a - \hat{\beta}_b) + \#\{b \succ a\} \log F(\hat{\beta}_b - \hat{\beta}_a)$$
$$< \#\{a \succ b\} \log F(\hat{\beta}_b - \hat{\beta}_a) + \#\{b \succ a\} \log F(\hat{\beta}_a - \hat{\beta}_b),$$

using the property on the counts in the dataset, the fact that $\hat{\beta}_a < \hat{\beta}_b$ and $F$ is strictly monotonic. Hence, using these expressions in Equation (6), we obtain

$$\mathcal{L}(\hat{\beta}) < \sum_{x,y \notin \{a,b\}} \#\{x \succ y\} \log F(\hat{\beta}_x - \hat{\beta}_y)$$

$$+ \sum_{y \notin \{a,b\}} \#\{a \succ y\} \log F(\hat{\beta}_b - \hat{\beta}_y) + \sum_{y \notin \{a,b\}} \#\{b \succ y\} \log F(\hat{\beta}_a - \hat{\beta}_y)$$

$$+ \sum_{x \notin \{a,b\}} \#\{x \succ b\} \log F(\hat{\beta}_x - \hat{\beta}_a) + \sum_{x \notin \{a,b\}} \#\{x \succ a\} \log F(\hat{\beta}_x - \hat{\beta}_b)$$

$$+ \#\{a \succ b\} \log F(\hat{\beta}_b - \hat{\beta}_a) + \#\{b \succ a\} \log F(\hat{\beta}_a - \hat{\beta}_b)$$

$$= \sum_{x,y \notin \{a,b\}} \#\{x \succ y\} \log F(\tilde{\beta}_x - \tilde{\beta}_y)$$

$$+ \sum_{y \notin \{a,b\}} \#\{a \succ y\} \log F(\tilde{\beta}_a - \tilde{\beta}_y) + \sum_{y \notin \{a,b\}} \#\{b \succ y\} \log F(\tilde{\beta}_b - \tilde{\beta}_y)$$

$$+ \sum_{x \notin \{a,b\}} \#\{x \succ b\} \log F(\tilde{\beta}_x - \tilde{\beta}_b) + \sum_{x \notin \{a,b\}} \#\{x \succ a\} \log F(\tilde{\beta}_x - \tilde{\beta}_a)$$

$$+ \#\{a \succ b\} \log F(\tilde{\beta}_a - \tilde{\beta}_b) + \#\{b \succ a\} \log F(\tilde{\beta}_b - \tilde{\beta}_a)$$

$$= \sum_{x \neq y} \#\{x \succ y\} \log F(\tilde{\beta}_x - \tilde{\beta}_y)$$

$$= \mathcal{L}(\tilde{\beta}),$$

implying that $\tilde{\beta}$ has a strictly higher log-likelihood than the MLE $\hat{\beta}$, leading to a contradiction. Therefore, every every MLE $\hat{\beta}$ must satisfy $\hat{\beta}_a \geq \hat{\beta}_b$ under this condition. $\qquad \square$

## D   Proof of Theorem 4.2

Let $\#$ and $\tilde{\#}$ be two datasets as defined in Definition 4.1, with (unique) MLEs $\hat{\beta}$ and $\tilde{\beta}$. That is, $\tilde{\#}$ is the same as $\#$, except with $\alpha > 0$ comparisons of $a \succ b$ added to it. We prove that for all alternatives $x \in \mathcal{X}$, we have

$$\tilde{\beta}_a - \tilde{\beta}_x \geq \hat{\beta}_a - \tilde{\beta}_x.$$

The proof for the $b$ part ($\tilde{\beta}_b - \tilde{\beta}_x \leq \hat{\beta}_b - \tilde{\beta}_x$) is completely symmetric.

Let the log-likelihood function with respect to $\#$ be denoted by $\mathcal{L}$, while the log-likelihood function with respect to $\tilde{\#}$ be denoted by $\tilde{\mathcal{L}}$. Any alternative could be set as the reference, but we use $a$ as the reference alternative in this proof for ease of exposition. As $\tilde{\#}$ is the same as $\#$, except with $\alpha$ additional $a \succ b$ comparisons, we have

$$\tilde{\mathcal{L}}(\beta) = \mathcal{L}(\beta) + \alpha \log F(\beta_a - \beta_b).$$

Let $\mathcal{U}$ denote the set of alternatives $u \in \mathcal{X} \setminus \{a\}$ for which $\tilde{\beta}_u - \tilde{\beta}_a \leq \hat{\beta}_u - \hat{\beta}_a$.[12] And, let $\mathcal{V}$ denote the set of alternatives $v \in \mathcal{X} \setminus \{a\}$ for which $\tilde{\beta}_v - \tilde{\beta}_a > \hat{\beta}_v - \hat{\beta}_a$. Our goal is to show that $\mathcal{U} = \mathcal{X} \setminus \{a\}$, or equivalently that $\mathcal{V} = \phi$.

First, we show that $b \in \mathcal{U}$. Suppose for the sake of contradiction, that $\tilde{\beta}_b - \tilde{\beta}_a > \hat{\beta}_b - \hat{\beta}_a$. Then, this implies that $\alpha \log F(\tilde{\beta}_a - \tilde{\beta}_b) < \alpha \log F(\hat{\beta}_a - \hat{\beta}_b)$ as both $\log$ and $F$ are strictly monotonic, and $\alpha > 0$. Further, as $\hat{\beta}$ maximizes $\mathcal{L}$, we have $\mathcal{L}(\tilde{\beta}) \leq \mathcal{L}(\hat{\beta})$. This implies that $\mathcal{L}(\tilde{\beta}) + \alpha \log F(\tilde{\beta}_a - \tilde{\beta}_b) < \mathcal{L}(\hat{\beta}) + \alpha \log F(\hat{\beta}_a - \hat{\beta}_b)$. Or, $\tilde{\mathcal{L}}(\tilde{\beta}) < \tilde{\mathcal{L}}(\hat{\beta})$, which is a contradiction as $\tilde{\beta}$ is the maximizer of $\tilde{\mathcal{L}}$. This proves that $\tilde{\beta}_b - \tilde{\beta}_a \leq \hat{\beta}_b - \hat{\beta}_a$, i.e. $b \in \mathcal{U}$.

Next, suppose for the sake of contradiction that $\mathcal{V} \neq \phi$. We can rewrite the log-likelihood function $\mathcal{L}(\beta)$ as

$$\mathcal{L}(\beta) = \sum_{(x,y) \in \mathcal{X}^2} \#\{x \succ y\} \log F(\beta_x - \beta_y)$$

$$= \sum_{\{x,y\} \subseteq \mathcal{X}} \left[ \#\{x \succ y\} \log F(\beta_x - \beta_y) + \#\{y \succ x\} \log F(\beta_y - \beta_x) \right],$$

where the latter summation is over unordered pairs of alternatives $\{x, y\}$ with $x \neq y$. Denote each term in this expression by $\ell_{xy}(\beta_x - \beta_y)$, i.e.

$$\ell_{xy}(\eta) = \#\{x \succ y\} \log F(\eta) + \#\{y \succ x\} \log F(-\eta).$$

As $F$ is log-concave, $\log F$ is a concave function. And since linear transformations, positive scalar multiplication and addition preserve concavity, each of these functions $\ell_{xy}$ is also concave.

Let us define operator $\Delta_{xy}$ to be such that when it is applied to a $\beta$ vector, it returns the difference in $\beta$ values of alternatives $x$ and $y$. That is, $\Delta_{xy}\beta := \beta_x - \beta_y$. Using this notation to rewrite the log-likelihood function, we have

$$\mathcal{L}(\beta) = \sum_{\{x,y\}} \ell_{xy}(\Delta_{xy}\beta)$$

$$= \sum_{u \in \mathcal{U}} \ell_{ua}(\Delta_{ua}\beta) + \sum_{v \in \mathcal{V}} \ell_{va}(\Delta_{va}\beta) + \sum_{\{u,p\} \subseteq \mathcal{U}} \ell_{up}(\Delta_{up}\beta)$$

$$+ \sum_{\{v,q\} \subseteq \mathcal{V}} \ell_{vq}(\Delta_{vq}\beta) + \sum_{(u,v) \in \mathcal{U} \times \mathcal{V}} \ell_{vu}(\Delta_{vu}\beta),$$

where the first two terms are the paired terms with $a$, the third is pairs within $\mathcal{U}$, the fourth is pairs within $\mathcal{V}$, and the last is for pairs across $\mathcal{U}$ and $\mathcal{V}$. For each $v \in \mathcal{V}$, we know $\tilde{\beta}_v - \tilde{\beta}_a > \hat{\beta}_v - \hat{\beta}_a$. Hence, we can write $\Delta_{va}\tilde{\beta} = \Delta_{va}\hat{\beta} + \delta_v$,[13] where $\delta_v > 0$ for each $v \in \mathcal{V}$. Recall, $\hat{\beta}$ is the maximizer of $\mathcal{L}$. Hence, $\mathcal{L}(\hat{\beta}_\mathcal{U}, \tilde{\beta}_\mathcal{V}) < \mathcal{L}(\hat{\beta}_\mathcal{U}, \hat{\beta}_\mathcal{V})$,[14] as the MLE $\hat{\beta}$ is unique, and $\mathcal{V} \neq \phi$. This implies that

$$\sum_{u \in \mathcal{U}} \ell_{ua}(\Delta_{ua}\hat{\beta}) + \sum_{v \in \mathcal{V}} \ell_{va}(\Delta_{va}\tilde{\beta}) + \sum_{\{u,p\} \subseteq \mathcal{U}} \ell_{up}(\Delta_{up}\hat{\beta}) + \sum_{\{v,q\} \subseteq \mathcal{V}} \ell_{vq}(\Delta_{vq}\tilde{\beta}) + \sum_{(u,v) \in \mathcal{U} \times \mathcal{V}} \ell_{vu}(\tilde{\beta}_v - \hat{\beta}_u)$$

$$< \sum_{u \in \mathcal{U}} \ell_{ua}(\Delta_{ua}\hat{\beta}) + \sum_{v \in \mathcal{V}} \ell_{va}(\Delta_{va}\hat{\beta}) + \sum_{\{u,p\} \subseteq \mathcal{U}} \ell_{up}(\Delta_{up}\hat{\beta}) + \sum_{\{v,q\} \subseteq \mathcal{V}} \ell_{vq}(\Delta_{vq}\hat{\beta}) + \sum_{(u,v) \in \mathcal{U} \times \mathcal{V}} \ell_{vu}(\hat{\beta}_v - \hat{\beta}_u).$$

Cancelling terms that appear on both sides (because of the same $\hat{\beta}_\mathcal{U}$), and plugging in $\Delta_{va}\tilde{\beta} = \Delta_{va}\hat{\beta} + \delta_v$, we have

$$\sum_{v \in \mathcal{V}} \ell_{va}(\Delta_{va}\hat{\beta} + \delta_v) + \sum_{\{v,q\} \subseteq \mathcal{V}} \ell_{vq}(\Delta_{vq}\hat{\beta} + \delta_v - \delta_q) + \sum_{(u,v) \in \mathcal{U} \times \mathcal{V}} \ell_{vu}(\Delta_{vu}\hat{\beta} + \delta_v)$$

$$< \sum_{v \in \mathcal{V}} \ell_{va}(\Delta_{va}\hat{\beta}) + \sum_{\{v,q\} \subseteq \mathcal{V}} \ell_{vq}(\Delta_{vq}\hat{\beta}) + \sum_{(u,v) \in \mathcal{U} \times \mathcal{V}} \ell_{vu}(\Delta_{vu}\hat{\beta}).$$

Or in other words,

$$\sum_{(u,v) \in \mathcal{U} \times \mathcal{V}} \ell_{vu}(\Delta_{vu}\hat{\beta} + \delta_v) - \sum_{(u,v) \in \mathcal{U} \times \mathcal{V}} \ell_{vu}(\Delta_{vu}\hat{\beta}) \tag{7}$$

$$< - \left[ \sum_{v \in \mathcal{V}} \ell_{va}(\Delta_{va}\hat{\beta} + \delta_v) + \sum_{\{v,q\} \subseteq \mathcal{V}} \ell_{vq}(\Delta_{vq}\hat{\beta} + \delta_v - \delta_q) - \sum_{v \in \mathcal{V}} \ell_{va}(\Delta_{va}\hat{\beta}) - \sum_{\{v,q\} \subseteq \mathcal{V}} \ell_{vq}(\Delta_{vq}\hat{\beta}) \right].$$

Intuitively, it says that if you increase each $\hat{\beta}_v$ by their $\delta_v$, the increase in likelihood because of the cross terms $\ell_{vu}$ is less than the loss because of the exclusive $v$ terms (or vice versa, i.e. the loss in likelihood because of $\ell_{vu}$ is higher than the increase because of the exclusive $v$ terms).

For each $u \in \mathcal{U}$, we know $\tilde{\beta}_u - \tilde{\beta}_a \leq \hat{\beta}_u - \hat{\beta}_a$. Hence, we can write $\Delta_{ua}\tilde{\beta} = \Delta_{ua}\hat{\beta} - \lambda_u$,[15] where $\lambda_u \geq 0$ for each $u \in \mathcal{U}$. We now compare $\tilde{\mathcal{L}}(\tilde{\beta}_{\mathcal{U}}, \tilde{\beta}_{\mathcal{V}})$ and $\tilde{\mathcal{L}}(\tilde{\beta}_{\mathcal{U}}, \hat{\beta}_{\mathcal{V}})$.[16] In other words, we compare

$$\sum_{u \in \mathcal{U}} \ell_{ua}(\Delta_{ua}\tilde{\beta}) + \sum_{v \in \mathcal{V}} \ell_{va}(\Delta_{va}\tilde{\beta}) + \sum_{\{u,p\} \subseteq \mathcal{U}} \ell_{up}(\Delta_{up}\tilde{\beta})$$

$$+ \sum_{\{v,q\} \subseteq \mathcal{V}} \ell_{vq}(\Delta_{vq}\tilde{\beta}) + \sum_{(u,v) \in \mathcal{U} \times \mathcal{V}} \ell_{vu}(\Delta_{vu}\tilde{\beta}) + \alpha \log F(\Delta_{ab}\tilde{\beta})$$

$$vs$$

$$\sum_{u \in \mathcal{U}} \ell_{ua}(\Delta_{ua}\tilde{\beta}) + \sum_{v \in \mathcal{V}} \ell_{va}(\Delta_{va}\hat{\beta}) + \sum_{\{u,p\} \subseteq \mathcal{U}} \ell_{up}(\Delta_{up}\tilde{\beta})$$

$$+ \sum_{\{v,q\} \subseteq \mathcal{V}} \ell_{vq}(\Delta_{vq}\hat{\beta}) + \sum_{(u,v) \in \mathcal{U} \times \mathcal{V}} \ell_{vu}(\hat{\beta}_v - \tilde{\beta}_u) + \alpha \log F(\Delta_{ab}\tilde{\beta}).$$

Note that the last term $\alpha \log F(\Delta_{ab}\beta)$ appears with a $\tilde{\beta}$ in both the equations because we know $b \in \mathcal{U}$. Cancelling terms that appear on both sides (because of the same $\tilde{\beta}_{\mathcal{U}}$), and plugging in $\Delta_{va}\tilde{\beta} = \Delta_{va}\hat{\beta} + \delta_v$ for $v \in \mathcal{V}$, as well as $\Delta_{ua}\tilde{\beta} = \Delta_{ua}\hat{\beta} - \lambda_u$ for $u \in \mathcal{U}$, we are comparing

$$\sum_{v \in \mathcal{V}} \ell_{va}(\Delta_{va}\hat{\beta} + \delta_v) + \sum_{\{v,q\} \subseteq \mathcal{V}} \ell_{vq}(\Delta_{vq}\hat{\beta} + \delta_v - \delta_q) + \sum_{(u,v) \in \mathcal{U} \times \mathcal{V}} \ell_{vu}(\Delta_{vu}\hat{\beta} + \lambda_u + \delta_v)$$

$$vs$$

$$\sum_{v \in \mathcal{V}} \ell_{va}(\Delta_{va}\hat{\beta}) + \sum_{\{v,q\} \subseteq \mathcal{V}} \ell_{vq}(\Delta_{vq}\hat{\beta}) + \sum_{(u,v) \in \mathcal{U} \times \mathcal{V}} \ell_{vu}(\Delta_{vu}\hat{\beta} + \lambda_u).$$

And, rearranging this, we compare

$$\sum_{(u,v) \in \mathcal{U} \times \mathcal{V}} \ell_{vu}(\Delta_{vu}\hat{\beta} + \lambda_u + \delta_v) - \sum_{(u,v) \in \mathcal{U} \times \mathcal{V}} \ell_{vu}(\Delta_{vu}\hat{\beta} + \lambda_u)$$

$$vs \tag{8}$$

$$-\left[ \sum_{v \in \mathcal{V}} \ell_{va}(\Delta_{va}\hat{\beta} + \delta_v) + \sum_{\{v,q\} \subseteq \mathcal{V}} \ell_{vq}(\Delta_{vq}\hat{\beta} + \delta_v - \delta_q) - \sum_{v \in \mathcal{V}} \ell_{va}(\Delta_{va}\hat{\beta}) - \sum_{\{v,q\} \subseteq \mathcal{V}} \ell_{vq}(\Delta_{vq}\hat{\beta}) \right].$$

If $\lambda_u$ were zero, we know that the left hand side (i.e. the equation placed above in (8)) is smaller (than the one placed below) because of equation (7). But, we now show that this holds even for $\lambda_u \geq 0$ by concavity of the functions $\ell_{vu}$. For each $(u,v) \in \mathcal{U} \times \mathcal{V}$, we can write

$$\ell_{vu}(\Delta_{vu}\hat{\beta} + \lambda_u + \delta_v) - \ell_{vu}(\Delta_{vu}\hat{\beta} + \lambda_u) = \int_{\Delta_{vu}\hat{\beta} + \lambda_u}^{\Delta_{vu}\hat{\beta} + \lambda_u + \delta_v} \ell'_{vu}(t)dt,$$

where $\ell'_{vu}$ is the derivative of $\ell_{vu}$.[17] Changing the variable of intergration,

$$\ell_{vu}(\Delta_{vu}\hat{\beta} + \lambda_u + \delta_v) - \ell_{vu}(\Delta_{vu}\hat{\beta} + \lambda_u) = \int_{\Delta_{vu}\hat{\beta}}^{\Delta_{vu}\hat{\beta} + \delta_v} \ell'_{vu}(s + \lambda_u)ds.$$

But, we know that $\ell_{vu}$ is a concave function, implying that $\ell'_{vu}$ is monotonically decreasing. Hence, $\ell'_{vu}(s + \lambda_u) \leq \ell'_{vu}(s)$ for every $s$, as $\lambda_u \geq 0$. This gives us

$$\ell_{vu}(\Delta_{vu}\hat{\beta} + \lambda_u + \delta_v) - \ell_{vu}(\Delta_{vu}\hat{\beta} + \lambda_u) = \int_{\Delta_{vu}\hat{\beta}}^{\Delta_{vu}\hat{\beta}+\delta_v} \ell'_{vu}(s + \lambda_u)ds$$

$$\leq \int_{\Delta_{vu}\hat{\beta}}^{\Delta_{vu}\hat{\beta}+\delta_v} \ell'_{vu}(s)ds$$

$$= \ell_{vu}(\Delta_{vu}\hat{\beta} + \delta_v) - \ell_{vu}(\Delta_{vu}\hat{\beta}).$$

Taking a summation of the left hand side over all $(u, v) \in \mathcal{U} \times \mathcal{V}$, shows that this summation is less than or equal to the left hand side of Equation (7). Hence, this summation is strictly smaller than the right hand side of Equation (7) (because of Equation (7) itself). This in turn implies that the equation placed above in (8) is strictly smaller than the one placed below. In other words, $\tilde{\mathcal{L}}(\tilde{\beta}) < \tilde{\mathcal{L}}(\tilde{\beta}_{\mathcal{U}}, \hat{\beta}_{\mathcal{V}})$, contradicting the fact that $\tilde{\beta}$ is the maximizer of $\tilde{\mathcal{L}}$. Hence, $V = \phi$. In other words, for each $x \in \mathcal{X} \setminus \{a\}$, $\tilde{\beta}_x - \tilde{\beta}_a \leq \hat{\beta}_x - \hat{\beta}_a$. $\qquad \square$

# E  Proof of Theorem 5.3

In this proof, we will consider datasets in which the frequency of pairwise comparisons need not be integral. However, we can choose the frequencies to be rational and then obtain a result for integral frequencies by multiplying by a common denominator.

Consider $\mathcal{X} = \{a, b, c\}$, and let the dataset be as follows. $\#\{a \succ b\} = 5 + \epsilon$, $\#\{b \succ a\} = 5$, $\#\{a \succ c\} = 5 + \epsilon$, $\#\{c \succ a\} = 5$, $\#\{b \succ c\} = 100$ and $\#\{c \succ b\} = 1$. Here, $\epsilon$ is a constant lying in $[0, 1]$. Observe that for any $\epsilon > 0$, this dataset conforms to Definition 5.1 if we label $x_1, x_2, x_3 = a, b, c$. To show violation of PMC, we show that there exists $\epsilon_o \in (0, 1]$ for which the (unique) MLE $\hat{\beta}$ violates the corresponding requirement of $\hat{\beta}_a \geq \hat{\beta}_b \geq \hat{\beta}_c$.

The log-likelihood function for this data is given by

$$\mathcal{L}_\epsilon(\beta) = (5 + \epsilon) \log F(\beta_a - \beta_b) + 5 \log F(\beta_b - \beta_a) + 100 \log F(\beta_b - \beta_c) + \log F(\beta_c - \beta_b)$$
$$+ (5 + \epsilon) \log F(\beta_a - \beta_c) + 5 \log F(\beta_c - \beta_a).$$

Observe that every alternative has been compared with every other alternative, and hence, the comparison graph $\mathcal{G}_\#$ is strongly connected. Further, as $F$ is strictly monotonic, continuous and strictly log-concave, the MLE exists and is unique for any $\epsilon \in [0, 1]$ (by Lemmas 2.1 and 2.4). Further, the log-likelihood $\mathcal{L}_\epsilon(\beta)$ is a strictly concave function (for each $\epsilon \in [0, 1]$). Any alternative could be set as the reference, but we use $c$ as the reference alternative in this proof for ease of exposition. That is, our domain is $\mathcal{D} = \{\beta \in \mathbb{R}^\mathcal{X} : \beta_c = 0\}$. The (unique) maximum likelihood estimator is given by

$$\hat{\beta}(\epsilon) = \underset{\beta \in \mathcal{D}}{\operatorname{argmax}} \ \mathcal{L}_\epsilon(\beta).$$

We first show that $\hat{\beta}(\epsilon)$ is a continuous function of $\epsilon$. As $F$ is strictly monotonic and continuous, for each $\epsilon \in [0, 1]$, Lemma 2.3 tells us that the MLE is bounded as

$$\|\hat{\beta}(\epsilon)\|_\infty \leq |\mathcal{X}| \cdot \max_{(x,y) \in \mathcal{X}^2} \delta_\epsilon(x, y),$$

where $\delta_\epsilon$ is the perfect-fit distance, but is now dependent on $\epsilon$. For the dataset at hand, these perfect-fit distances are given by

$$\delta_\epsilon(a, b) = F^{-1}\left(\frac{5 + \epsilon}{10 + \epsilon}\right), \quad \delta_\epsilon(b, c) = F^{-1}\left(\frac{100}{101}\right) \quad \text{and} \quad \delta_\epsilon(a, c) = F^{-1}\left(\frac{5 + \epsilon}{10 + \epsilon}\right).$$

And, $\delta_\epsilon(b, a) = -\delta_\epsilon(a, b), \delta_\epsilon(c, b) = -\delta_\epsilon(b, c)$ and $\delta_\epsilon(c, a) = -\delta_\epsilon(a, c)$, as $F^{-1}(1 - x) = -F^{-1}(x)$. Further, the first three distances are non-negative (making the remaining three non-positive) as $F^{-1}(x) \geq 0$ for $x \geq \frac{1}{2}$. Hence, the bound on the MLE simplifies to

$$\|\hat{\beta}(\epsilon)\|_\infty \leq 3 \cdot \max\left(F^{-1}\left(\frac{5 + \epsilon}{10 + \epsilon}\right), F^{-1}\left(\frac{100}{101}\right)\right).$$

As $F$ is strictly monotonic, it implies that $F^{-1}$ is also strictly increasing. Applying this, we have

$$F^{-1}\left(\frac{5+\epsilon}{10+\epsilon}\right) \leq F^{-1}\left(\frac{6}{11}\right) < F^{-1}\left(\frac{100}{101}\right),$$

as $\epsilon \in [0,1]$. Therefore, the bound on the MLE further simplifies to

$$\|\hat{\beta}(\epsilon)\|_\infty \leq 3\, F^{-1}\left(\frac{100}{101}\right),$$

for any $\epsilon \in [0,1]$. Hence, the MLE optimization problem can be rewritten as

$$\hat{\beta}(\epsilon) = \underset{\beta \in \mathcal{D}:\|\beta\|_\infty \leq 3\, F^{-1}\left(\frac{100}{101}\right)}{\operatorname{argmax}} \mathcal{L}_\epsilon(\beta).$$

This shows that we are optimizing over a compact space. Hence, by the Theorem of the Maximum [Berge, 1963, Jehle and Reny, 2011], both the maximum likelihood and the corresponding maximizer $\hat{\beta}(\epsilon)$ are continuous in the parameter $\epsilon$, for all $\epsilon \in [0,1]$.

Next, we analyze the MLE at $\epsilon = 0$. The log-likelihood function for this value of $\epsilon$ is

$$\mathcal{L}_0(\beta) = 5\log F(\beta_a - \beta_b) + 5\log F(\beta_b - \beta_a) + 100\log F(\beta_b - \beta_c) + \log F(\beta_c - \beta_b)$$
$$+ 5\log F(\beta_a - \beta_c) + 5\log F(\beta_c - \beta_a). \tag{9}$$

For ease of exposition, we use $\beta^\dagger$ to denote the MLE when $\epsilon = 0$, i.e.

$$\beta^\dagger := \hat{\beta}(0) = \underset{\beta \in \mathcal{D}}{\operatorname{argmax}} \mathcal{L}_0(\beta).$$

Recall that we used $c$ as the reference alternative, and hence, $\beta_c^\dagger = 0$. Our goal is to show that $\beta_b^\dagger > \beta_a^\dagger > \beta_c^\dagger$. To this end, we first show that $\beta_a^\dagger = \beta_b^\dagger/2$, i.e. in terms of $\beta$ values, $a$ lies at the mid-point of $b$ and $c$. Suppose for the sake of contradiction that $\beta_a^\dagger \neq \beta_b^\dagger/2$. Consider another vector $\tilde{\beta} \in \mathcal{D}$ that is the same as $\beta^\dagger$, except with the distances between $\beta$ values of $b$ & $a$ and $a$ & $c$ swapped. This can be achieved by setting

$$\tilde{\beta}_x = \begin{cases} \beta_x^\dagger & ;\ \text{if } x \in \{b, c\} \\ \beta_b^\dagger - \beta_a^\dagger & ;\ \text{if } x = a. \end{cases}$$

Then, we have $\tilde{\beta}_a - \tilde{\beta}_c = \beta_b^\dagger - \beta_a^\dagger$ and $\tilde{\beta}_b - \tilde{\beta}_a = \beta_a^\dagger - \beta_c^\dagger$. Hence, the log-likelihood at this point is given by

$$\mathcal{L}_0(\tilde{\beta}) = 5\log F(\tilde{\beta}_a - \tilde{\beta}_b) + 5\log F(\tilde{\beta}_b - \tilde{\beta}_a) + 100\log F(\tilde{\beta}_b - \tilde{\beta}_c) + \log F(\tilde{\beta}_c - \tilde{\beta}_b)$$
$$+ 5\log F(\tilde{\beta}_a - \tilde{\beta}_c) + 5\log F(\tilde{\beta}_c - \tilde{\beta}_a)$$
$$= 5\log F(\beta_c^\dagger - \beta_a^\dagger) + 5\log F(\beta_a^\dagger - \beta_c^\dagger) + 100\log F(\beta_b^\dagger - \beta_c^\dagger) + \log F(\beta_c^\dagger - \beta_b^\dagger)$$
$$+ 5\log F(\beta_b^\dagger - \beta_a^\dagger) + 5\log F(\beta_a^\dagger - \beta_b^\dagger)$$
$$= \mathcal{L}_0(\beta^\dagger).$$

That is, swapping these distances does not change the likelihood, because of symmetry. Now, consider a new vector $\bar{\beta} = (\beta^\dagger + \tilde{\beta})/2$. Note that, as $\beta_a^\dagger \neq \beta_b^\dagger/2$, it implies that $\beta_a^\dagger \neq \beta_b^\dagger - \beta_a^\dagger = \tilde{\beta}_a$. In other words, $\tilde{\beta} \neq \beta^\dagger$. Therefore, applying strict concavity of $\mathcal{L}_0$, we have

$$\mathcal{L}_0(\bar{\beta}) = \mathcal{L}_0\left(\frac{\beta^\dagger + \tilde{\beta}}{2}\right) > \frac{\mathcal{L}_0(\beta^\dagger) + \mathcal{L}_0(\tilde{\beta})}{2} = \mathcal{L}_0(\beta^\dagger),$$

which is a contradiction as $\beta^\dagger$ is the maximizer of $\mathcal{L}_0$. This proves that $\beta_a^\dagger = \beta_b^\dagger/2$. In other words, $\beta^\dagger$ is of the form $(\beta_b^\dagger/2, \beta_b^\dagger, 0)$. Hence, $\beta^\dagger$ continues to be the maximizer of $\mathcal{L}_0$ among the vectors $\mathcal{A} = \{(\alpha/2, \alpha, 0) : \alpha \in \mathbb{R}\} \subseteq \mathcal{D}$. Rewriting the log-likelihood (9) for vectors in $\mathcal{A}$, we have

$$\mathcal{L}_0((\alpha/2, \alpha, 0)) = 5\log F\left(-\frac{\alpha}{2}\right) + 5\log F\left(\frac{\alpha}{2}\right) + 100\log F(\alpha) + \log F(-\alpha)$$
$$+ 5\log F\left(\frac{\alpha}{2}\right) + 5\log F\left(-\frac{\alpha}{2}\right)$$
$$= 10\log F\left(\frac{\alpha}{2}\right) + 10\log F\left(-\frac{\alpha}{2}\right) + 100\log F(\alpha) + \log F(-\alpha).$$

Overloading notation, we denote this log-likelihood by $\mathcal{L}_0(\alpha)$, and this is maximized at $\alpha = \beta_b^\dagger$. For ease of exposition, denote the composition of $\log$ and $F$ by $G$, i.e. $G := \log F$. As $F$ is strictly monotonic, differentiable and strictly log-concave, $G$ is also strictly monotonic and differentiable, and is strictly concave.[18] Rewriting the log-likelihood with this notation, we have

$$\mathcal{L}_0(\alpha) = 10G\left(\frac{\alpha}{2}\right) + 10G\left(-\frac{\alpha}{2}\right) + 100G\left(\alpha\right) + G\left(-\alpha\right).$$

We show that this function is not maximized at any $\alpha \leq 0$. In other words, $\beta_b^\dagger > 0$. Computing the derivative of $\mathcal{L}_0$, we have

$$\mathcal{L}_0'(\alpha) = 5G'\left(\frac{\alpha}{2}\right) - 5G'\left(-\frac{\alpha}{2}\right) + 100G'\left(\alpha\right) - G'\left(-\alpha\right).$$

As $G$ is strictly concave, it implies that $G'$ is strictly decreasing. Hence, for $\alpha \leq 0$, it implies that $G'(\frac{\alpha}{2}) \geq G'(-\frac{\alpha}{2})$ and $G'(\alpha) \geq G'(-\alpha)$. This shows that for $\alpha \leq 0$, we have

$$\mathcal{L}_0'(\alpha) \geq 99G'\left(\alpha\right) > 0,$$

where the last inequality holds as $G$ is a strictly increasing function, leading to $G'$ being positive.[19] In other words, if $\alpha \leq 0$, the log-likelihood can be strictly increased by taking an infinitesimally small step in the direction $\left[\frac{1}{2}, 1, 0\right]$. Hence, none of these points maximizes $\mathcal{L}_0$, and $\beta_b^\dagger > 0$. Also, recall that $\beta^\dagger$ was of the form $(\beta_b^\dagger/2, \beta_b^\dagger, 0)$; this proves that $\beta_b^\dagger > \beta_a^\dagger > \beta_c^\dagger$.

Finally, recall that we need $\epsilon > 0$ for the dataset to conform to Definition 5.1 with the labelling $x_1, x_2, x_3, = a, b, c$. To be able to find such a value of $\epsilon$, we use continuity of $\hat{\beta}(\epsilon)$. By continuity, we know that for every $\gamma > 0$, there exists $\delta > 0$ such that $\|\hat{\beta}(\epsilon) - \hat{\beta}(0)\|_\infty < \gamma$ for all $|\epsilon - 0| < \delta$. Define $\theta := \beta_b^\dagger - \beta_a^\dagger > 0$. Then, choose $\gamma = \theta/3$, and let $\delta_o$ denote the corresponding value of $\delta$. Hence, choose $\epsilon_o = \min(\delta_o/2, 1) > 0$. For this value of $\epsilon_o$, we indeed have $\|\hat{\beta}(\epsilon_o) - \hat{\beta}(0)\|_\infty < \theta/3$. That is,

$$\hat{\beta}(\epsilon_o)_b > \beta_b^\dagger - \frac{\theta}{3} \qquad \text{and} \qquad \hat{\beta}(\epsilon_o)_a < \beta_a^\dagger + \frac{\theta}{3}.$$

Hence,

$$\hat{\beta}(\epsilon_o)_b - \hat{\beta}(\epsilon_o)_a > \beta_b^\dagger - \beta_a^\dagger - \frac{2\theta}{3} = \theta - \frac{2\theta}{3} > 0.$$

Therefore, at $\epsilon = \epsilon_o \in (0, 1]$, the MLE satisfies $\hat{\beta}(\epsilon_o)_b > \hat{\beta}(\epsilon_o)_a$. Hence, the dataset with $\epsilon = \epsilon_o$ satisfies the PMC condition, but the corresponding MLE does not conform to the corresponding ordering, proving violation of pairwise majority consistency. $\qquad\square$

## F  Proof of Theorem 6.3

Consider $\mathcal{X} = \{a, b, c\}$, and let the two datasets be as follows. The first dataset is such that $\#^1\{a \succ c\} = 5 + \epsilon, \#^1\{c \succ a\} = 5 - \epsilon, \#^1\{c \succ b\} = 100, \#^1\{b \succ c\} = 1$, and has zero counts otherwise. The second dataset is such that $\#^2\{a \succ c\} = 5 + \epsilon, \#^2\{c \succ a\} = 5 - \epsilon, \#^2\{b \succ a\} = 100, \#^2\{a \succ b\} = 1$, and has zero counts otherwise. Here, $\epsilon$ is a constant lying in $(0, 1]$.

First, we analyze the MLE for the dataset $\#^1$. As the comparison graph $\mathcal{G}_{\#^1}$ is strongly connected, and $F$ is strictly monotonic, continuous and strictly log-concave, the MLE $\hat{\beta}^1$ exists and is unique (by Lemmas 2.1 and 2.4). Further, the pair $(a, c)$ satisfies the condition of Lemma 2.2, similarly does the pair $(b, c)$. Applying the lemma for the pair $(a, c)$ says that the MLE satisfies

$$\hat{\beta}_a^1 = \hat{\beta}_c^1 + F^{-1}\left(\frac{5 + \epsilon}{10}\right) > \hat{\beta}_c^1,$$

as $(5 + \epsilon)/10$ is larger than $1/2$. Similarly, applying Lemma 2.2 for the pair $(b, c)$ says that the MLE satisfies

$$\hat{\beta}_b^1 = \hat{\beta}_c^1 + F^{-1}\left(\frac{1}{1 + 100}\right) < \hat{\beta}_c^1,$$

as $1/101$ is smaller than $1/2$. Putting these equations together, we have $\hat{\beta}_a^1 > \hat{\beta}_c^1 > \hat{\beta}_b^1$.

Next, we analyze the MLE for the dataset $\#^2$. As the comparison graph $\mathcal{G}_{\#^2}$ is strongly connected, and $F$ is strictly monotonic, continuous and strictly log-concave, the MLE $\hat{\beta}^2$ exists and is unique (by Lemmas 2.1 and 2.4). Further, the pair $(c, a)$ satisfies the condition of Lemma 2.2, similarly does the pair $(b, a)$. Applying the lemma for the pair $(c, a)$ says that the MLE satisfies

$$\hat{\beta}_c^2 = \hat{\beta}_a^2 + F^{-1}\left(\frac{5 - \epsilon}{10}\right) < \hat{\beta}_a^2,$$

as $(5 - \epsilon)/10$ is smaller than $1/2$. Similarly, applying Lemma 2.2 for the pair $(b, a)$ says that the MLE satisfies

$$\hat{\beta}_b^2 = \hat{\beta}_a^2 + F^{-1}\left(\frac{100}{1 + 100}\right) > \hat{\beta}_a^2,$$

as $100/101$ is larger than $1/2$. Putting these equations together, we have $\hat{\beta}_b^2 > \hat{\beta}_a^2 > \hat{\beta}_c^2$. Hence, both datasets $\#^1$ and $\#^2$ have MLEs $\hat{\beta}^1$ and $\hat{\beta}^2$ such that $\hat{\beta}_a^1 > \hat{\beta}_c^1$ and $\hat{\beta}_a^2 > \hat{\beta}_c^2$.

Finally, we analyze the MLE for the dataset $\# = \#^1 + \#^2$ obtained by pooling both datasets $\#^1$ and $\#^2$. Recall that the proof so far holds for any constant $\epsilon \in (0, 1]$; but, from this point on, we allow $\epsilon$ to take the value of zero as well, i.e. $\epsilon \in [0, 1]$. The log-likelihood function for the pooled data $\#$ is given by

$$\mathcal{L}_\epsilon(\beta) = 100 \log F(\beta_c - \beta_b) + \log F(\beta_b - \beta_c) + 100 \log F(\beta_b - \beta_a) + \log F(\beta_a - \beta_b)$$
$$+ (10 + 2\epsilon) \log F(\beta_a - \beta_c) + (10 - 2\epsilon) \log F(\beta_c - \beta_a).$$

Observe that every alternative has been compared with every other alternative, and hence, the comparison graph $\mathcal{G}_\#$ is strongly connected. Further, as $F$ is strictly monotonic, continuous and strictly log-concave, the MLE exists and is unique for any $\epsilon \in [0, 1]$ (by Lemmas 2.1 and 2.4). Further, the log-likelihood $\mathcal{L}_\epsilon(\beta)$ is a strictly concave function (for each $\epsilon \in [0, 1]$). Any alternative could be set as the reference, but we use $a$ as the reference alternative in this proof for ease of exposition. That is, our domain is $\mathcal{D} = \{\beta \in \mathbb{R}^{\mathcal{X}} : \beta_a = 0\}$. The (unique) maximum likelihood estimator is given by

$$\hat{\beta}(\epsilon) = \underset{\beta \in \mathcal{D}}{\mathrm{argmax}}\ \mathcal{L}_\epsilon(\beta).$$

Similar to the proof of Theorem 5.3, we first show that $\hat{\beta}(\epsilon)$ is a continuous function of $\epsilon$. As $F$ is strictly monotonic and continuous, for each $\epsilon \in [0, 1]$, Lemma 2.3 tells us that the MLE is bounded as

$$\|\hat{\beta}(\epsilon)\|_\infty \leq |\mathcal{X}| \cdot \max_{(x,y) \in \mathcal{X}^2} \delta_\epsilon(x, y),$$

where $\delta_\epsilon$ is the perfect-fit distance, but is now dependent on $\epsilon$. For our pooled dataset, these perfect-fit distances are given by

$$\delta_\epsilon(b, a) = F^{-1}\left(\frac{100}{101}\right), \quad \delta_\epsilon(c, b) = F^{-1}\left(\frac{100}{101}\right) \quad \text{and} \quad \delta_\epsilon(a, c) = F^{-1}\left(\frac{10 + 2\epsilon}{20}\right).$$

And, $\delta_\epsilon(a, b) = -\delta_\epsilon(b, a), \delta_\epsilon(b, c) = -\delta_\epsilon(c, b)$ and $\delta_\epsilon(c, a) = -\delta_\epsilon(a, c)$, as $F^{-1}(1 - x) = -F^{-1}(x)$. Further, the first three distances are non-negative (making the remaining three non-positive) as $F^{-1}(x) \geq 0$ for $x \geq \frac{1}{2}$. Hence, the bound on the MLE simplifies to

$$\|\hat{\beta}(\epsilon)\|_\infty \leq 3 \cdot \max\left(F^{-1}\left(\frac{100}{101}\right), F^{-1}\left(\frac{10 + 2\epsilon}{20}\right)\right).$$

As $F$ is strictly monotonic, it implies that $F^{-1}$ is also strictly increasing. Applying this, we have

$$F^{-1}\left(\frac{10 + 2\epsilon}{20}\right) \leq F^{-1}\left(\frac{12}{20}\right) < F^{-1}\left(\frac{100}{101}\right),$$

as $\epsilon \in [0, 1]$. Therefore, the bound on the MLE further simplifies to

$$\|\hat{\beta}(\epsilon)\|_\infty \leq 3 \, F^{-1}\left(\frac{100}{101}\right),$$

for any $\epsilon \in [0, 1]$. Hence, the MLE optimization problem can be rewritten as

$$\hat{\beta}(\epsilon) = \underset{\beta \in \mathcal{D}: \|\beta\|_\infty \leq 3 \, F^{-1}\left(\frac{100}{101}\right)}{\operatorname{argmax}} \mathcal{L}_\epsilon(\beta).$$

This shows that we are optimizing over a compact space. Hence, by the Theorem of the Maximum, both the maximum likelihood and the corresponding maximizer $\hat{\beta}(\epsilon)$ are continuous in the parameter $\epsilon$, for all $\epsilon \in [0, 1]$.

Next, we analyze the MLE at $\epsilon = 0$. The log-likelihood function for this value of $\epsilon$ is

$$\begin{aligned}
\mathcal{L}_0(\beta) = {} & 100 \log F(\beta_c - \beta_b) + \log F(\beta_b - \beta_c) + 100 \log F(\beta_b - \beta_a) + \log F(\beta_a - \beta_b) \\
& + 10 \log F(\beta_a - \beta_c) + 10 \log F(\beta_c - \beta_a).
\end{aligned} \tag{10}$$

For ease of exposition, we use $\beta^\dagger$ to denote the MLE when $\epsilon = 0$, i.e.

$$\beta^\dagger := \hat{\beta}(0) = \underset{\beta \in \mathcal{D}}{\operatorname{argmax}} \, \mathcal{L}_\epsilon(\beta).$$

Recall that we used $a$ as the reference alternative, and hence, $\beta_a^\dagger = 0$. Our goal is to show that $\beta_c^\dagger > \beta_b^\dagger > \beta_a^\dagger$. To this end, we first show that $\beta_b^\dagger = \beta_c^\dagger/2$, i.e. in terms of $\beta$ values, $b$ lies at the mid-point of $c$ and $a$. Suppose for the sake of contradiction that $\beta_b^\dagger \neq \beta_c^\dagger/2$. Consider another vector $\tilde{\beta} \in \mathcal{D}$ that is the same as $\beta^\dagger$, except with the distances between $c$ & $b$ and $b$ & $a$ swapped. This can be achieved by setting

$$\tilde{\beta}_x = \begin{cases} \beta_x^\dagger & ; \text{ if } x \neq \{a, c\} \\ \beta_c^\dagger - \beta_b^\dagger & ; \text{ if } x = b. \end{cases}$$

Then, we have $\tilde{\beta}_b - \tilde{\beta}_a = \beta_c^\dagger - \beta_b^\dagger$ and $\tilde{\beta}_c - \tilde{\beta}_b = \beta_b^\dagger - \beta_a^\dagger$. Hence, the likelihood at this point is given by

$$\begin{aligned}
\mathcal{L}_0(\tilde{\beta}) = {} & 100 \log F(\tilde{\beta}_c - \tilde{\beta}_b) + \log F(\tilde{\beta}_b - \tilde{\beta}_c) + 100 \log F(\tilde{\beta}_b - \tilde{\beta}_a) + \log F(\tilde{\beta}_a - \tilde{\beta}_b) \\
& + 10 \log F(\tilde{\beta}_a - \tilde{\beta}_c) + 10 \log F(\tilde{\beta}_c - \tilde{\beta}_a) \\
= {} & 100 \log F(\beta_b^\dagger - \beta_a^\dagger) + \log F(\beta_a^\dagger - \beta_b^\dagger) + 100 \log F(\beta_c^\dagger - \beta_b^\dagger) + \log F(\beta_b^\dagger - \beta_c^\dagger) \\
& + 10 \log F(\beta_a^\dagger - \beta_c^\dagger) + 10 \log F(\beta_c^\dagger - \beta_a^\dagger) \\
= {} & \mathcal{L}_0(\beta^\dagger).
\end{aligned}$$

That is, swapping these distances does not change the likelihood, because of symmetry. Now, consider a new vector $\bar{\beta} = (\beta^\dagger + \tilde{\beta})/2$. Note that, as $\beta_b^\dagger \neq \beta_c^\dagger/2$, it implies that $\beta_b^\dagger \neq \beta_c^\dagger - \beta_b^\dagger = \tilde{\beta}_b$. In other words, $\tilde{\beta} \neq \beta^\dagger$. Therefore, applying strict concavity of $\mathcal{L}_0$, we have

$$\mathcal{L}_0(\bar{\beta}) = \mathcal{L}_0\left(\frac{\beta^\dagger + \tilde{\beta}}{2}\right) > \frac{\mathcal{L}_0(\beta^\dagger) + \mathcal{L}_0(\tilde{\beta})}{2} = \mathcal{L}_0(\beta^\dagger),$$

which is a contradiction as $\beta^\dagger$ is the maximizer of $\mathcal{L}_0$. This proves that $\beta_b^\dagger = \beta_c^\dagger/2$. In other words, $\beta^\dagger$ is of the form $(0, \beta_c^\dagger/2, \beta_c^\dagger)$. Hence, $\beta^\dagger$ continues to be the maximizer of $\mathcal{L}_0$ among the vectors $\mathcal{A} = \{(0, \alpha/2, \alpha) : \alpha \in \mathbb{R}\} \subseteq \mathcal{D}$. Rewriting the log-likelihood (10) for vectors in $\mathcal{A}$, we have

$$\begin{aligned}
\mathcal{L}_0((0, \alpha/2, \alpha)) = {} & 100 \log F\left(\frac{\alpha}{2}\right) + \log F\left(-\frac{\alpha}{2}\right) + 100 \log F\left(\frac{\alpha}{2}\right) + \log F\left(-\frac{\alpha}{2}\right) \\
& + 10 \log F(-\alpha) + 10 \log F(\alpha) \\
= {} & 200 \log F\left(\frac{\alpha}{2}\right) + 2 \log F\left(-\frac{\alpha}{2}\right) + 10 \log F(\alpha) + 10 \log F(-\alpha).
\end{aligned}$$

Overloading notation, we denote this log-likelihood by $\mathcal{L}_0(\alpha)$, and this is maximized at $\alpha = \beta_c^\dagger$. For ease of exposition, denote the composition of $\log$ and $F$ by $G$, i.e. $G := \log F$. As $F$ is strictly monotonic, differentiable and strictly log-concave, $G$ is also strictly monotonic and differentiable, and is strictly concave.[20] Rewriting the log-likelihood with this notation, we have

$$\mathcal{L}_0(\alpha) = 200G\left(\frac{\alpha}{2}\right) + 2G\left(-\frac{\alpha}{2}\right) + 10G(\alpha) + 10G(-\alpha).$$

We show that this function is not maximized at any $\alpha \leq 0$. In other words, $\beta_c^\dagger > 0$. Computing the derivative of $\mathcal{L}_0$, we have

$$\mathcal{L}_0'(\alpha) = 100G'\left(\frac{\alpha}{2}\right) - G'\left(-\frac{\alpha}{2}\right) + 10G'(\alpha) - 10G'(-\alpha).$$

As $G$ is strictly concave, it implies that $G'$ is strictly decreasing. Hence, for $\alpha \leq 0$, it implies that $G'(\frac{\alpha}{2}) \geq G'(-\frac{\alpha}{2})$ and $G'(\alpha) \geq G'(-\alpha)$. This shows that for $\alpha \leq 0$, we have

$$\mathcal{L}_0'(\alpha) \geq 99G'\left(\frac{\alpha}{2}\right) > 0,$$

where the last inequality holds as $G$ is a strictly increasing function, leading to $G'$ being positive.[21] In other words, if $\alpha \leq 0$, the log-likelihood can be strictly increased by taking an infinitesimally small step in the direction $\left[0, \frac{1}{2}, 1\right]$. Hence, none of these points maximizes $\mathcal{L}_0$, and $\beta_c^\dagger > 0$. Also, recall that $\beta^\dagger$ was of the form $(0, \beta_c^\dagger/2, \beta_c^\dagger)$; this proves that $\beta_c^\dagger > \beta_b^\dagger > \beta_a^\dagger$.

Finally, recall that the initial part of the proof (analyzing the MLE for the individual datasets) works only for $0 < \epsilon \leq 1$. Hence, we need to use an $\epsilon$ value strictly larger than zero even for the pooled dataset. To be able to find such a value of $\epsilon$, we use continuity of $\hat{\beta}(\epsilon)$. By continuity, we know that for every $\gamma > 0$, there exists $\delta > 0$ such that $\|\hat{\beta}(\epsilon) - \hat{\beta}(0)\|_\infty < \gamma$ for all $|\epsilon - 0| < \delta$. Define $\theta := \beta_c^\dagger - \beta_a^\dagger > 0$. Then, choose $\gamma = \theta/3$, and let $\delta_o$ denote the corresponding value of $\delta$. Hence, choose $\epsilon_o = \min(\delta_o/2, 1) > 0$. For this value of $\epsilon_o$, we indeed have $\|\hat{\beta}(\epsilon_o) - \hat{\beta}(0)\|_\infty < \theta/3$. That is,

$$\hat{\beta}(\epsilon_o)_c > \beta_c^\dagger - \frac{\theta}{3} \qquad \text{and} \qquad \hat{\beta}(\epsilon_o)_a < \beta_a^\dagger + \frac{\theta}{3}.$$

Hence,

$$\hat{\beta}(\epsilon_o)_c - \hat{\beta}(\epsilon_o)_a > \beta_c^\dagger - \beta_a^\dagger - \frac{2\theta}{3} = \theta - \frac{2\theta}{3} > 0.$$

Therefore, at $\epsilon = \epsilon_o \in (0, 1]$, the MLE (on the pooled data) satisfies $\hat{\beta}(\epsilon_o)_c > \hat{\beta}(\epsilon_o)_a$. Hence, for $\epsilon = \epsilon_o$, the two datasets $\#^1$ and $\#^2$ have MLEs $\hat{\beta}^1$ and $\hat{\beta}^2$ such that $\hat{\beta}_a^1 > \hat{\beta}_c^1$ and $\hat{\beta}_a^2 > \hat{\beta}_c^2$, but the MLE $\hat{\beta}$ on the pooled dataset $\# = \#^1 + \#^2$ satisfies $\hat{\beta}_a < \hat{\beta}_c$, proving violation of separability. $\square$

## Footnotes

[6]That is, $K$ denotes the total number of comparisons in the dataset, and $\eta$ denotes the smallest positive comparison number in it (or equivalently, the smallest positive weight in $\mathcal{G}_\#$). Also note that, $F^{-1}$ exists in $(0, 1)$ as $F$ is strictly monotonic and continous.

[7]assuming all alternatives of $C_t$ are placed on the real line according to their $\beta$ values.

[8] In the case when $S$ has the reference alternative $r$, the exact effect can be achieved by instead decreasing the beta values of all alternatives in $\mathcal{X} \setminus S$ by the same constant.

[9] In the case when $a$ was set as the reference, we could always perform this optimization by placing the reference on some other alternative, and then shifting the complete learned vector back such that $a$ is the reference again. Observe that this does not affect the learned distance of $(\hat{\beta}_a - \hat{\beta}_b)$, for which we are proving the desired property.

[10] assuming all the alternatives are placed on the real line according to their $\beta$ values.

[11]In case either $a$ or $b$ is the reference alternative, shift $\tilde\beta$ after swapping these two alternatives' utilities such that the reference is restored. Rest of the proof remains the same as the shifted beta vector has the same likelihood as the unshifted one.

[12]Even though $\tilde{\beta}_a = \hat{\beta}_a = 0$ as $a$ is the reference, we do not omit it in some parts for better clarity.

[13]Equivalently, this could be written as $\tilde{\beta}_v = \hat{\beta}_v + \delta_v$, as $\tilde{\beta}_a = \hat{\beta}_a = 0$.

[14]Recall that alternative $a$ has been set as the reference, and hence it zero in both these terms.

[15]Equivalently, this could be written as $\tilde{\beta}_u = \hat{\beta}_u - \lambda_u$, as $\tilde{\beta}_a = \hat{\beta}_a = 0$.

[16]That is, we are again keeping the $\mathcal{U}$ part fixed, while changing the $\mathcal{V}$ part from $\tilde{\beta}_{\mathcal{V}}$ to $\hat{\beta}_{\mathcal{V}}$.

[17]which exists, as $F$ is differentiable.

[18]This part of the proof does not require concavity of $G$, but we use it nevertheless as it simplifies the proof.

[19]Strictly speaking, a function might be strictly increasing and have a derivative that is not strictly positive at every point (in particular, the derivative might be zero at stationary points). But in our case, as $G'$ is also a strictly decreasing function, it cannot be zero at any point, because that would make it negative at larger points, violating strict monotonicity of $G$.

[20]This part of the proof does not require concavity of $G$, but we use it nevertheless as it simplifies the proof.

[21]Strictly speaking, a function might be strictly increasing and have a derivative that is not strictly positive at every point (in particular, the derivative might be zero at stationary points). But in our case, as $G'$ is also a strictly decreasing function, it cannot be zero at any point, because that would make it negative at larger points, violating strict monotonicity of $G$.