[Reviews · NeurIPS 2020]

Review 1

Summary and Contributions: The authors analyze the structure of the MLE of random utility models for comparison data relative to several natural axioms from social choice theory. The motivation behind these axioms is driven either by purely common sense, i.e., what would be the intuitive properties of a learned MLE ranking, or by "rules" from social choice theory. Specifically, the focus on four axioms: Pareto efficiency, monotonicity, pairwise majority consistency and separability, and for each they test if it is satisfied by an ML ranking. The authors consider a general class of random utility models that are based on probability functions F with CDF-like properties, which in particularly includes the common Thurstone–Mosteller Bradley–Terry models. The authors prove four theorems for each of the axioms. They show that the MLE agrees with the first two axioms if F satisfies certain properties. For majority consistency and separability, the authors demonstrate counter examples where MLE fails to follow these axioms.

Strengths: The paper gives some nice insights about the MLE for a large class of random utility models for pairwise comparison data. Particularly, the main result (Theorem 4.2) is very helpful in confirming that only a few natural assumptions on F are needed in order for MLE to satisfy a general monotonicity relative to any third alternative, although the result is pretty intuitive. The paper also highlights important problems associated with an MLE that could arise if used in real world decision-making scenarios.

Weaknesses: - I failed to see the connection between the classic economical definition of Pareto-efficiency and the definition proposed by the authors (lines 167-168 and Def. 3.1). Moreover, in Def. 3.1. the constraints "#{a>x} > #{b>x} _and_ ${x>a} < #{x>b} for _every_ alternative x" seem to be too strict (probably, this was done in order to simplify the analysis of Thm. 3.2). But it does not match the "intuitive" explanation (lines 167-168). For example, consider the case of three alternatives {a,b,x}, if we have the following pairs #{a>b}=100, #{b>a}=1, #{a>x}=2, #{b>x}=1, #{x>a}=100, #{x>b}=99. This example violates the definition of Pareto efficiency, however it is clear that a should be ranked higher than b: x>a>b. - Lines 231-242: The assumption that with high probability one can expect the increase of the number of comparisons {a>x} for _all_ x (in order to have \beta_a increased) does not seem to be very realistic. Why would pairs {a > x} for all x suddenly appear? - Although theorems 5.3 and 6.3 state interesting results, they are only proved for the 3-alternatives case. Will the result hold when there are more than 3 alternatives?

Correctness: The proofs seem to be correct, however, note my questions below.

Clarity: The paper is well-written for the most part, however Figure 1 needs to be redone. Why there are there only 12 numbers, while there are supposed to be 16 (4^2)? Giving only numbers #{x>y} without explicitly saying which pair (x,y) it corresponds to, is making it hard to understand the relationship between the alternatives. Very nice Figure 2 though.

Relation to Prior Work: Yes.

Reproducibility: Yes

Additional Feedback: - Line 181: "in the multi-agent case the dataset satisfies Def. 3.1 whp", is this a provable claim? - For both sections 5 and 6, it would be interesting to see what is the % of violated datasets (either experimentally or theoretically). - I only glanced through the proofs in the appendix, but one issue puzzled me: in the Proof of Thm 5.3: why is \epsilon in [0,1] perturbing the counts? Is this a continuous relaxation of counts? Same question for Thm 6


Review 2

Summary and Contributions: The paper takes an axiomatic view of the relationship between pairwise comparisons and the ranking of alternatives output by the MLE of a RUM trained on those comparisons, finding that they obey monotonicity and Pareto efficiency while failing to uphold PMC and separability.

Strengths: This paper does an excellent job providing a general view of the benefits and costs of using standard MLE with RUMs to estimate preferences from pairwise comparisons. Four axioms are stated and explained, two are shown to hold for all RUMs, and the other two are shown not to hold under very mild conditions of the RUM's iid noise CDF. The axiomatic approach here is very relevant in the current research climate, understanding the pros and cons of using these models for social choice is

Weaknesses: I find few weaknesses with this work, as it mostly just leaves me wanting to see extensions to e.g. the non-pairwise case or to more axioms. For the Monotonicity and PMC axioms, I really liked the visualization of datasets that violate the axioms. While I agree that your counterexample datasets are not pathological, you also point out there may be evidence that a RUM is not a good model for the data. It'd be interesting to know if there are RUMs that still generate data violating these properties, and to see say the parameters of a Bradley-Terry model such that P(violation) is high for a few data points. To be clear- I don't think the paper needs this to warrant publication and am simply curious because the work is so interesting.

Correctness: Everything is correct as far as I can tell. I skimmed the proofs in the appendix but did not make deep dives.

Clarity: The paper is very well written.

Relation to Prior Work: Overall, yes. I have only one nit: In one of the footnotes, rankings are mentioned as a special case of pairwise comparisons, which stuck me as odd- both are special cases of partial rankings but I don't see a way to frame one as contained within the other. I certainly don't think that this work handles ranking data as a special case of pairwise comparisons, as the extension of RUMs to ranking data iirc will cause many of these F(x-y) terms to become many-dimensional integrals, which I assume complicates the proofs at least some. That said, I still the past work is different from this, as it focuses on ranking data that comes from e.g. the Mallows model, and even considering the pairwise comparisons from those rankings as data is probably not going to give you data that comes from a RUM.

Reproducibility: Yes

Additional Feedback: It would have been nice to have examples of pairwise comparison aggregation methods that satisfy the two properties (PMC and seperability) that RUM MLEs do not. One thing I'm left wondering is whether there are any ranking aggregation methods that take all these pairwise comparisons as input and output a ranking that satisfies all these properties, perhaps replacing \beta_i with the number of items ranked below i so that the numerical axioms relying on the RUM params still mean something. Would be interested in knowing whether this failure to meet all four axioms is a drawback of just the MLE of RUMs or whether there's something more general going on under the hood. Update: After reviewing the author response and discussing with the other reviewers, my score remains the same.


Review 3

Summary and Contributions: This paper performs a normative analysis for MLE in the setting where the aggregation rule takes the pairwise comparisons as input. In detail, under a class of random utility models, this work checks several normative axioms for MLE and finds that MLE satisfies a weaker version of Pareto efficiency: if alternative a dominates alternative b and other alternatives do not hurt this dominance, MLE will output a dominates b as well. Monotonicity: adding a>b comparisons will help a and hurt b in the results of MLE. This work also presents two counterexamples to show that MLE does not respect `` majority will’’ and has a problem when combining two datasets. The main insight of this work is that unlike the normative approach, the statistical methods rely crucially on the underlying model assumptions. The normative approach may be robust by definition. However, with the statistical methods, to fit the underlying model, the change of a single pairwise comparison (e.g. the number of (a>b)) may affect the parameters of other alternatives a lot. This somewhat leads to the non-robustness of the statistical methods. Therefore, for MLE, several axioms (e.g. majority will) may not be satisfied in general and some axioms (e.g. monotonicity) require some subtle properties of the underlying model.

Strengths: The problem this work considers is elegant and important. This work mentions the practical use of the statistical method, MLE, in the situation of kidney exchange, which is a good motivation for this work. This work also points out that the statistical method can be much more accurate when the underlying model is right but can violate several basic axioms (e.g. fails to satisfy majority will) in other situations. This trade-off is interesting and formally analyzed in this work. The normative analysis of statistical methods is a good aspect. The standard analysis framework presented in this work can be useful to the community. Moreover, this paper is well-written. After reading this paper, the reason that why MLE fails to satisfy some axioms in some situations is pretty clear. I think this work provides a good insight into the comparison between the normative approach and the statistical approach.

Weaknesses: One weakness of this paper is that it only analyzes the existed approach and the analysis somewhat follows a standard way. The log-concave property is interesting and it would be nice if this work can give more intuition in this part (e.g. shows why monotonicity fails with a non-log-concave model). The idea of performing normative analysis on statistical methods is proposed in previous works. Moreover, this work is not that closely related to the NeurIPS community. But overall I like this work and I think it provides several useful insights.

Correctness: Yes

Clarity: Yes

Relation to Prior Work: Yes

Reproducibility: Yes

Additional Feedback: -------------- Thanks for the rebuttal. After reading the rebuttal and other reviewer's reviews, I am still leaning torward acceptance.


Review 4

Summary and Contributions: The paper studies social choice axioms that iid random utility models (RUM) do/don't satisfy. By considering conditions on F, the difference CDF for the RUM, some axioms are satisfied when conditions on F are met. A notion of Pareto efficiency and of monotonicity are found to be broadly satisfied, while notions of pairwise majority consistency and separability are found to not be satisfied, by counterexample. Update: I appreciate the author rebuttal on several fronts, both in response to my points about the Ford condition and on "participation incentive". I have raised my score from 4 to 6.

Strengths: 1. Building connections between social choice/voting and discrete choice/random utility theory is an important bridging of closely related but not often conversant literatures. 2. The paper is generally well-written and easy to understand/follow.

Weaknesses: 1. Some details of the definitions and assumptions of the theorems need to be ironed out, see detailed comments. 2. The notion of "monotonicity" is not quite a notion of "participation incentive" as the paper claims, see discussion.

Correctness: Mostly correct. See nit about how the definitions and theorems sew together.

Clarity: Yes.

Relation to Prior Work: It's not clear why the authors chose to write separate existence and uniqueness conditions without any discussion of the "Ford condition" (Ford, 1957) or work since then that builds on the condition. It's clear that the authors generalize the Ford notion, giving conditions on F and bounds on the inf-norm. I do not know if these specific versions (e.g. inf-norm bounds) exist in the literature, but I am willing to assume they are new. But this work on existence and uniqueness could be better connected to prior work.

Reproducibility: Yes

Additional Feedback: -Some part of Sec 2.1 and Sec 2.2, presenting Lemmas 2.1-2.3, should reference the well-known Ford condition [1] for existence and uniqueness, which is essentially an assumption that the directed pairwise comparison graph be strongly connected. It’s not clear when existence (Lemma 2.1) and uniqueness (Lemma 2.3) will be employed separately, and it’s quite clear that the conditions for existence and uniqueness combine to require strong connectivity (plus the conditions on F in this generalization of the problem considered by Ford), and so I’m wondering if it isn’t cleaner (and less work) to simply present a common condition for existence and uniqueness? [1] Ford Jr LR (1957) Solution of a ranking problem from binary comparisons. Amer. Math. Monthly 64(8):28–33. - Definition 3.1 could probably be weakened by looking only at the ratio (or difference) of a>b vs b>a. Consider a dataset where a>b many times, b>a once. And then there are 100,000 comparisons between b and x_1, 90,000 of which are x>b. But only 100 comparisons of a and x_1, 99 of which are a>x_1. Then the “If” part of Pareto efficiency isn’t satisfied, even if this scenario would clearly give \hat \beta_a \ge \hat \beta_b. Maybe working with conditions on differences/ratios is really hard; if that’s the case, an explanation would be helpful. - Theorem 3.2 doesn’t require the existence or uniqueness of the MLE. As such, it is easy to construct examples (a beats b, a beats all x, all x beat b) where the MLE doesn’t exist, and thus the claim of Pareto efficiency doesn’t hold since \hat \beta doesn’t exist. - L200 “participation incentive”; actually, this idea of “incentive" is not something delivered by monotonicity. The incentives for participation in pairwise comparisons depends on the expected value of the change in score when competing. It is well-known in games that use Elo ratings, which are based on a version of random utility modeling, there can be incentive problems if the noise distribution of the model doesn’t match the noise distribution of the game. Consider a game where player A has an Elo rating of 2000 and player B has a rating of 100. The model predicts A will beat B. But maybe the model assumes a logistic difference but the game is actually more random than that, based on some small element of luck. In that was, the tail probability will be miscalibrated, and there’s a much larger chance of B beating A than the model predicts. Player A would lose a lot then. Player A’s expected value from competing would be negative. All I’m flagging here is: monotonicity, while nice, does not go all the way to establishing a participation incentive. That ultimately requires a mess of details related to private information about noise distributions, and to the best of this reviewers knowledge is a hard open problem. - The way Def 4.1 is written, assuming two unique datasets with unique MLEs, and Then “monotonicity requires that…”, it’s less clear than it ought to be that this “requirement” is on the model underlying the likelihood, essentially conditions on F (later given in Theorem 4.2). Try to make this definition clearer, possibly by invoking F as part of the definition. - While an “only if” version of Theorem 4.2 would be hard/a mess (because small local pathological changes in F could violate e.g. log-concavity but still have an MLE that satisfies monotonicity), it would be interesting to see/understand an example where the conditions on F aren’t met, and then the MLE does not satisfy monotonicity. Which is to say, somehow probe the necessity of these conditions.

[Author Response · NeurIPS 2020]

# Author Response: Axioms for Learning from Pairwise Comparisons #11464

**Review 1**

Regarding the definition of Pareto optimality: Please refer to the explanation in lines 173–177, which is based on the idea of pooling different RUM-produced datasets. This context motivated our definition, and in this model the demanding restriction on $a$ vs $x$ pairs is appropriate. (Your example makes sense, and is reminiscent of PMC.)

Your question about lines 231–242: Note that we assume (line 238) that the number of comparisons is uniform across all pairs, so we will see many $\#_i$-comparisons of $a$ vs $x$ for every $x$. Since we assume that the difference $\beta_a - \beta_x$ has increased, we will (whp) see more comparisons that go $a > x$.

Theorems 5.3 and 6.3 for more than three alternatives: crucially, the proofs of these theorems construct universal counterexamples, and these can easily be extended to more than three alternatives. We will mention this.

"only 12 numbers": this is because we don't need counts for $x$ vs $x$ comparisons. We'll clarify. The claim of line 181 is indeed provable. Regarding $\varepsilon$-perturbations, we were using a continuous relaxation, but by choosing rational $\varepsilon$ and blowing the counts up, we can achieve integral counts.

**Review 2**

On the existence of parameters leading to high probability of PMC violations: Our proof sketch in lines 181–183 suggests that if datasets are generated by a RUM, it satisfies the Pareto condition of Definition 3.1 for all pairs, and hence the dataset is very unlikely to lead to a PMC violation.

Regarding examples of other aggregation rules that satisfy PMC and separability: this is a good point. A simple example of such a rule is the one where for each alternative $x \in \mathcal{X}$, we let $\hat{\beta}_x$ be the number of comparisons where $x$ wins minus the number where it loses. (In a sense, this is a generalization of Borda's rule.) This rule satisfies Pareto, monotonicity, and separability, but still fails PMC (due to Example 5.2). In fact, separability and PMC are incompatible since one can prove that all separable aggregators are variants of this counting rule. Overall, we don't think this rule is promising: for instance, on dataset $\#^1$ of Example 6.2, it places $c$ above $a$, which seems wrong as we argue in lines 291–292. Thus, in the applications we have in mind, RUMs seem more appropriate than this simple counting scheme, also because RUMs will have more predictive accuracy.

**Review 3**

Regarding necessity of log-concavity assumptions (also raised in Review 4): indeed it would be desirable to know more about this. As Reviewer 4 notes, this is probably a difficult project, since uniqueness of the MLE is tricky to ascertain, and since the MLE cannot be computed anymore by straightforwardly solving a convex program.

**Review 4**

Thank you for pointing that we seem to have rediscovered conditions for existence and uniqueness, and that (for Bradley–Terry) these were already known by Ford (1957). Searching further, it turns out that Ford himself rediscovered the conditions, which were already contained in Zermelo (1928). It was separately pointed out to us that the conditions for general RUMs were stated in an ICML-18 paper by Zhao and Xia. In the revised version, we will be careful to reference the appropriate literature. Please note, though, that we did not view these results as contributions of the paper — that is why we deliberately put them in the model section. We just need them as lemmas for our main theorems about the axioms. For those proofs, we also need the more explicit (and apparently novel) bounds using the inf-norm.

Regarding combining the conditions about existence and uniqueness into one: in some places (chiefly for Pareto optimality), we only need existence. For Pareto optimality, as you note, we were not completely clear about existence requirements. We will say that the statement on line 180 of Definition 3.1 is conditioned on the existence of $\hat{\beta}$.

Regarding your proposed generalization of the Pareto definition: This is an interesting suggestion. It is possible that a condition like this can be satisfied if the MLE first normalizes counts so that all pairs are seen equally often. However, such a normalization might be a disadvantage on other examples.

Regarding whether monotonicity is a participation incentive: There are two separate strategic models at play here. In the context that you describe, the alternatives are strategic players (such as competitors in a sports tournament) and need to decide whether to engage in another game with an uncertain outcome. You are right that in this model, monotonicity does not necessarily give a participation incentive. However, we were motivated by a different model. We think of the problem as aggregating preferences or opinions from voters, and then a participation incentive must assure voters that if they report additional pairwise comparisons, then the aggregate will move to be more aligned with the voter's opinion. Monotonicity, as we define it, gives exactly this guarantee.

Regarding the uniqueness requirement in Definition 4.1: we include a uniqueness requirement here so it makes sense to compare the positions of alternatives on different datasets. In general, the definition makes sense applied to aggregators that are not always unique. To keep it general, we prefer not to refer to the function $F$ in the definition of an axiom.

[Meta-Review · NeurIPS 2020]

The authors adequately addressed the major concerns of the reviewers in their response, and we are happy to recommend acceptance. We strongly encourage the authors to consider the points raised in the reviews as they prepare the final version of the paper.